# Certifying Stability of Reinforcement Learning Policies using Generalized Lyapunov Functions

**Kehan Long**    **Jorge Cortés**    **Nikolay Atanasov**
Contextual Robotics Institute
University of California San Diego
{k3long, cortes, natanasov}@ucsd.edu

## Abstract

Establishing stability certificates for closed-loop systems under reinforcement learning (RL) policies is essential to move beyond empirical performance and offer guarantees of system behavior. Classical Lyapunov methods require a strict stepwise decrease in the Lyapunov function but such certificates are difficult to construct for learned policies. The RL value function is a natural candidate but it is not well understood how it can be adapted for this purpose. To gain intuition, we first study the linear quadratic regulator (LQR) problem and make two key observations. First, a Lyapunov function can be obtained from the value function of an LQR policy by augmenting it with a residual term related to the system dynamics and stage cost. Second, the classical Lyapunov decrease requirement can be relaxed to a generalized Lyapunov condition requiring only decrease on average over multiple time steps. Using this intuition, we consider the nonlinear setting and formulate an approach to learn generalized Lyapunov functions by augmenting RL value functions with neural network residual terms. Our approach successfully certifies the stability of RL policies trained on Gymnasium and DeepMind Control benchmarks. We also extend our method to jointly train neural controllers and stability certificates using a multi-step Lyapunov loss, resulting in larger certified inner approximations of the region of attraction compared to the classical Lyapunov approach. Overall, our formulation enables stability certification for a broad class of systems with learned policies by making certificates easier to construct, thereby bridging classical control theory and modern learning-based methods.

## 1   Introduction

Designing and certifying stable control policies is a core challenge in both control theory and reinforcement learning (RL). Although RL methods are capable of learning policies that optimize long-term performance, their stability guarantees are often lacking. The classical Lyapunov theory [Lyapunov, 1992] offers a principled framework for certifying stability but constructing a function that satisfies the required stepwise decrease condition is often challenging in practice. While value functions from optimal control or RL encode long-term cumulative cost, they do not naturally satisfy the Lyapunov stability conditions.

We propose certifying stability using a generalized notion of Lyapunov function [Fürnsinn et al., 2023, 2025] that replaces the stepwise decrease condition with a multi-step, weighted decrease criterion, allowing temporary increases over a finite horizon. As a result, constructing a generalized Lyapunov stability certificate is easier, enabling stability verification for a broader class of policies and systems.

For linear systems with quadratic cost under discounted optimal control, the value function typically fails to satisfy classical Lyapunov conditions due to the influence of the discount factor. Postoyan et al. [2017] show that augmenting the value function with a quadratic residual term allows certifying

stability via a linear matrix inequality (LMI), although the guarantees are conservative with respect to the discount factor. Leveraging this insight, we show that augmenting the value function with a quadratic residual leads to a valid generalized Lyapunov function, which certifies stability for a boarder range of discount factors.

Building on the success of our approach in the linear setting, we extend it to nonlinear systems, where classical Lyapunov analysis is substantially more difficult. We augment value functions learned by RL algorithms (e.g., PPO [Schulman et al., 2017], SAC [Haarnoja et al., 2018]) with a neural residual term to form generalized Lyapunov functions, and jointly learn state-dependent multi-step weights to satisfy the generalized decrease condition. This joint learning scheme enables certification of nonlinear RL policies in settings where classical Lyapunov functions are challenging to construct by learning generalized certificates that tolerate non-monotonic behavior along trajectories.

In addition to showing stability of fixed policies, we extend our approach to support the joint synthesis of stable neural controllers and generalized Lyapunov certificates. Prior works [Chang et al., 2019, Wu et al., 2023, Yang et al., 2024] frame this problem as learning a controller–certificate pair that maximizes the volume of a certifiable region, while iteratively training on counter-examples where Lyapunov conditions fail. We incorporate our generalized multi-step condition into this formulation and show that it enables larger formally verifiable inner approximations of the region of attraction compared to standard one-step Lyapunov training.

**Contributions.** In summary, we make the following contributions:

- We introduce an approach for certifying stability of reinforcement learning policies by constructing generalized Lyapunov functions, composed of the policy's value function and a learned neural residual. Empirically, our formulation certifies policies on Gymnasium [Towers et al., 2024] and DeepMind Control Suite [Tassa et al., 2018] benchmarks where classical single-step Lyapunov methods fail.
- We show analytically that, for the linear quadratic regulator (LQR) problem, the validity of the proposed generalized Lyapunov function can be certified by a set of LMIs.
- We extend the formulation to jointly learn a control policy and a stability certificate using a generalized multi-step Lyapunov loss, resulting in larger certified regions of attraction compared to classical Lyapunov approaches.
- We provide an open-source implementation of our method at `https://github.com/ExistentialRobotics/Generalized_Policy_Stability`.

**Related Work.** Optimal control [Bertsekas, 2012] and reinforcement learning (RL) [Sutton et al., 1998] offer a variety of principled techniques for designing control policies to optimize the performance of intelligent agents and dynamical systems. In the special case of linear systems with quadratic rewards, the optimality and stability of LQR policies [Anderson and Moore, 1990] are well understood. However, despite a variety of methods for synthesizing policies (e.g., PPO [Schulman et al., 2017], SAC [Haarnoja et al., 2018], and DDPG [Lillicrap et al., 2015] are among the most widely used) for nonlinear systems with nonlinear costs, very few techniques are available for certifying the stability of the resulting closed-loop system.

Several works have explored connections between Lyapunov stability [Lyapunov, 1992] and RL, mainly in the context of safe RL. Perkins and Barto [2002] proposed a safe RL formulation by learning to switch among hand-designed Lyapunov-stable controllers. Berkenkamp et al. [2017] developed a model-based RL algorithm that uses Gaussian process dynamics and Lyapunov verification to safely improve policies while guaranteeing stability with high probability. Chow et al. [2018] formulate a Lyapunov-based method for safe RL in constrained Markov decision processes, using linear constraints to ensure safety during training. Hejase and Ozguner [2023] regularized policy training by penalizing violations of a learned Lyapunov condition to encourage stability in autonomous driving. Mittal et al. [2020] learn a Lyapunov function from suboptimal demonstrations and incorporate it as a terminal cost in a one-step model predictive control formulation to improve stability.

In parallel, there has been growing interest in jointly learning policies and Lyapunov functions from data to obtain stability guarantees for nonlinear systems. Chang et al. [2019] trained Lyapunov certificates with formal verification via satisfiability modulo theories. Other works constrain neural architectures to satisfy Lyapunov conditions structurally [Gaby et al., 2022, Boffi et al., 2021], or apply mixed-integer programming [Dai et al., 2021]. Long et al. [2023] studied neural distributionally robust Lyapunov functions under additive uncertainty. Wu et al. [2023] learn provably stable controllers

for discrete-time nonlinear systems by jointly training a policy and Lyapunov function, with formal verification via mixed-integer linear programs. Yang et al. [2024] use gradient-guided falsification for training a controller-certificate pair, and apply formal verification via $\alpha, \beta$-CROWN [Wang et al., 2021] to certify stability post training. See also Dawson et al. [2023] for a comprehensive survey.

## 2 Problem Formulation

Consider the discrete-time system:

$$\mathbf{x}_{k+1} = \mathbf{f}(\mathbf{x}_k, \mathbf{u}_k), \quad \mathbf{x}_k \in \mathcal{X} \subseteq \mathbb{R}^n, \quad \mathbf{u}_k \in \mathcal{U} \subseteq \mathbb{R}^m, \tag{1}$$

where $\mathcal{X}$ is open and $\mathbf{f} : \mathcal{X} \times \mathcal{U} \to \mathcal{X}$ is locally Lipschitz. A control policy $\boldsymbol{\pi} : \mathcal{X} \to \mathcal{U}$ for the system can be obtained by solving an infinite-horizon discounted optimal control problem:

$$J_\gamma^*(\mathbf{x}_0) = \min_{\boldsymbol{\pi}} J_\gamma^{\boldsymbol{\pi}}(\mathbf{x}_0) := \sum_{k=0}^{\infty} \gamma^k \, \ell(\mathbf{x}_k, \boldsymbol{\pi}(\mathbf{x}_k)), \tag{2}$$
$$\text{s.t. } \mathbf{x}_{k+1} = \mathbf{f}(\mathbf{x}_k, \boldsymbol{\pi}(\mathbf{x}_k)), \quad \mathbf{x}_k \in \mathcal{X}, \quad \boldsymbol{\pi}(\mathbf{x}_k) \in \mathcal{U},$$

where $\gamma \in (0, 1)$ is a discount factor and $\ell(\mathbf{x}, \mathbf{u})$ is a stage cost (or reward in the case of maximization), specifying the performance criterion.

Optimal control and reinforcement learning methods usually solve a problem like (2) to obtain a policy $\boldsymbol{\pi}$ and associated value function $J_\gamma^{\boldsymbol{\pi}}$. We consider a deterministic problem in our theoretical development for simplicity but apply our approach to stochastic problems in the evaluation (Section 5). Note that, while reinforcement learning methods work with stochastic control policies during training, only the mode of the final trained policy is used at test time, allowing us to treat it as deterministic.

Given a policy $\boldsymbol{\pi}$, we are interested in certifying whether it stabilizes the closed-loop system:

$$\mathbf{x}_{k+1} = \mathbf{f}(\mathbf{x}_k, \boldsymbol{\pi}(\mathbf{x}_k)). \tag{3}$$

We assume $\boldsymbol{\pi}(\mathbf{0}_n) = \mathbf{0}_m, \mathbf{f}(\mathbf{0}_n, \mathbf{0}_m) = \mathbf{0}_n, \ell(\mathbf{0}_n, \mathbf{0}_m) = 0$, and $\mathbf{0}_n \in \mathcal{X}$, so that the origin is an equilibrium point of the system. Stability can be certified by identifying a Lyapunov function.

**Definition 2.1 (Lyapunov Function).** Consider the closed-loop system (3). A continuous function $V : \mathcal{X} \to \mathbb{R}_{\geq 0}$ is a *Lyapunov function* if it satisfies:

$$V(\mathbf{0}_n) = 0, \quad V(\mathbf{x}) > 0 \quad \forall \mathbf{x} \in \mathcal{X} \setminus \{\mathbf{0}_n\}, \tag{4a}$$

$$V\big(\mathbf{f}(\mathbf{x}, \boldsymbol{\pi}(\mathbf{x}))\big) - V(\mathbf{x}) < 0 \quad \forall \mathbf{x} \in \mathcal{X} \setminus \{\mathbf{0}_n\}. \tag{4b}$$

Lyapunov functions certify the asymptotic stability of the system, as stated in the following result.

**Theorem 2.2 (Asymptotic Stability via a Lyapunov Function [Khalil, 1996, Theorem 3.3]).** *If there exists a Lyapunov function $V$ as in Definition 2.1, then the origin $\mathbf{x} = \mathbf{0}_n$ is an asymptotically stable equilibrium of the system (3).*

When $\boldsymbol{\pi}$ arises from (2), the corresponding value function $J_\gamma^{\boldsymbol{\pi}}$ is also available. We are interested in whether $J_\gamma^{\boldsymbol{\pi}}$ can itself certify the stability of (3) or assist in constructing a certificate.

## 3 Lyapunov Stability Analysis for Linear Quadratic Problems

To understand the relationship between a discounted value function $J_\gamma^{\boldsymbol{\pi}}$ obtained from (2) and a Lyapunov function $V$ certifying stability of the closed-loop system (3), we first study a simple setting with a linear system and quadratic stage cost.

Consider the discrete-time linear system: $\mathbf{x}_{k+1} = \mathbf{A}\mathbf{x}_k + \mathbf{B}\mathbf{u}_k$, where $\mathbf{x}_k \in \mathbb{R}^n$, $\mathbf{u}_k \in \mathbb{R}^m$, and $\mathbf{A}$, $\mathbf{B}$ are known constant matrices.

Choosing the stage cost in (2) as $\ell(\mathbf{x}, \mathbf{u}) = \mathbf{x}^\top \mathbf{Q}\mathbf{x} + \mathbf{u}^\top \mathbf{R}\mathbf{u}$ with $\mathbf{Q} = \mathbf{C}^\top \mathbf{C} \succeq 0$ and $\mathbf{R} \succ 0$ leads to the discounted LQR problem [Anderson and Moore, 1990]. The optimal value function $J_\gamma^*(\mathbf{x}) = \mathbf{x}^\top \mathbf{P}_\gamma \mathbf{x}$ is quadratic, where $\mathbf{P}_\gamma$ solves the discounted Algebraic Riccati Equation (ARE):

$$\mathbf{P}_\gamma = \mathbf{A}^\top \big(\mathbf{P}_\gamma - \mathbf{P}_\gamma \mathbf{B}\big(\gamma \mathbf{B}^\top \mathbf{P}_\gamma \mathbf{B} + \mathbf{R}\big)^{-1} \mathbf{B}^\top \mathbf{P}_\gamma\big) \mathbf{A} + \mathbf{Q}. \tag{5}$$

The corresponding optimal feedback gain is $\mathbf{K}_\gamma^\star = -(\gamma \mathbf{B}^\top \mathbf{P}_\gamma \mathbf{B} + \mathbf{R})^{-1} \mathbf{B}^\top \mathbf{P}_\gamma \mathbf{A}$, which yields the policy $\boldsymbol{\pi}_\gamma^*(\mathbf{x}_k) = \mathbf{K}_\gamma^\star \mathbf{x}_k$. Under this policy, the closed-loop system evolves as

$$\mathbf{x}_{k+1} = \mathbf{F}_\gamma^* \mathbf{x}_k, \quad \text{where} \quad \mathbf{F}_\gamma^* = \mathbf{A} + \mathbf{B} \mathbf{K}_\gamma^\star. \tag{6}$$

Although $J_\gamma^*$ satisfies the discounted Bellman equation, it is well understood that it does not guarantee Lyapunov decrease due to $\gamma \in (0, 1)$ [De Farias and Van Roy, 2003]. Moreover, verifying stability by directly analyzing the eigenvalues of $\mathbf{F}_\gamma^*$ and solving the discounted ARE (5) is nontrivial due to its nonlinear dependence on $\gamma$. This motivates the idea [Postoyan et al., 2017] of constructing valid Lyapunov functions by augmenting $J_\gamma^*$ with a residual term. To formalize this, we first state standard assumptions that ensure the existence of a stabilizing policy and a well-defined value function.

**Assumption 3.1.** *The pair $(\mathbf{A}, \mathbf{B})$ is stabilizable and the pair $(\mathbf{A}, \mathbf{C})$ is detectable.*

The following result shows the existence of a Lyapunov certificate derived from the value function.

**Theorem 3.2** (**Lyapunov Stability via LMIs for Discounted LQR** [Postoyan et al., 2017])**.** *Suppose Assumption 3.1 holds. Consider an optimal policy $\boldsymbol{\pi}_\gamma^*$ and value function $J_\gamma^*$ obtained from the LQR problem with discount $\gamma$. Let $\mathbf{P}$ denote the solution to the undiscounted ARE in (5) (with $\gamma = 1$). If there exist symmetric positive definite matrices $\mathbf{S}_0, \mathbf{S}_1 \succ 0$ and scalars $\varpi, \alpha > 0$ such that:*

$$\begin{bmatrix} \mathbf{A}^\top \mathbf{S}_0 \mathbf{A} - \mathbf{S}_0 + \mathbf{S}_1 - \varpi \mathbf{Q} & \mathbf{A}^\top \mathbf{S}_0 \mathbf{B} \\ \mathbf{B}^\top \mathbf{S}_0 \mathbf{A} & \mathbf{B}^\top \mathbf{S}_0 \mathbf{B} - \varpi \mathbf{R} \end{bmatrix} \preceq 0, \quad \alpha \mathbf{P} \preceq \mathbf{S}_1, \tag{7}$$

*then the function $V(\mathbf{x}) := J_\gamma^*(\mathbf{x}) + \frac{1}{\varpi} \mathbf{x}^\top \mathbf{S}_0 \mathbf{x}$ is a Lyapunov function for the closed-loop system (6), and certifies global exponential stability for any discount factor satisfying $\gamma > \frac{\varpi}{\varpi + \alpha}$.*

Theorem 3.2 shows that while the discounted value function $J_\gamma^*$ may not satisfy the Lyapunov condition on its own, it can be modified into a valid certificate when $\gamma$ is sufficiently large. The required residual term and a computable lower bound on $\gamma$ are obtained by solving a set of LMIs. However, this bound is often conservative compared to the true stability threshold $\gamma^*$ obtained by analyzing the closed-loop dynamics via the discounted ARE, as illustrated in the following example.

**Example 3.3** ([Postoyan et al., 2017, Example 1])**.** *Consider the scalar system $x_{k+1} = 2x_k + u_k$ with stage cost $\ell(x, u) = x^2 + u^2$. The optimal policy is $u_k = K_\gamma^\star x_k$, where*

$$K_\gamma^\star = -2 \left( 1 + 2 \left( 5\gamma - 1 + \sqrt{(5\gamma - 1)^2 + 4\gamma} \right)^{-1} \right)^{-1}.$$

*The closed-loop multiplier is $F_\gamma^* = 2 + K_\gamma^\star$, and the origin is globally exponentially stable if and only if $|F_\gamma^*| < 1$, which is equivalent to $\gamma > \gamma^* = 1/3$. However, applying the LMIs from Theorem 3.2 yields a feasible solution for $\gamma > 0.8090$, which is significantly more conservative.*

Example 3.3 highlights the limitation of classical Lyapunov analysis and motivates the development of an alternative formulation with less conservative conditions for certifying stability.

## 4 Generalized Lyapunov Stability and Applications to Linear Systems

Building on the observation from Example 3.3, we introduce a generalized notion of Lyapunov stability [Fürnsinn et al., 2023, Definition 2.2] that relaxes the classical pointwise decrease condition.

**Definition 4.1** (**Generalized Lyapunov Function**)**.** Consider the closed-loop system in (3). A continuous function $V : \mathcal{X} \to \mathbb{R}_{\geq 0}$ is a *generalized Lyapunov function* if it satisfies (4a) and there exist an integer $M \in \mathbb{N}_{>0}$ and state-dependent non-negative weights $\sigma_1(\mathbf{x}), \ldots, \sigma_M(\mathbf{x})$ such that

$$\frac{1}{M} \sum_{i=1}^M \sigma_i(\mathbf{x}) \geq 1, \tag{8}$$

and, for any $\mathbf{x} \in \mathcal{X} \setminus \{\mathbf{0}_n\}$, the following generalized decrease condition holds:

$$\frac{1}{M} \left( \sum_{i=1}^M \sigma_i(\mathbf{x}) V(\mathbf{x}_i) \right) - V(\mathbf{x}) < 0, \tag{9}$$

where $\mathbf{x}_{i+1} = \mathbf{f}(\mathbf{x}_i, \boldsymbol{\pi}(\mathbf{x}_i))$ with $\mathbf{x}_0 = \mathbf{x}$.

Condition (9) generalizes the decrease requirement in (4b) by allowing temporary increases in $V$ over individual steps, provided that its weighted average decreases over a finite horizon of length $M$. We state the stability guarantee provided by a generalized Lyapunov function in the next theorem.

**Theorem 4.2** (**Asymptotic Stability via a Generalized Lyapunov Function**). *Consider the closed-loop system in (3), and assume the policy $\boldsymbol{\pi}$ is Lipschitz on $\mathcal{X}$. If there exists a generalized Lyapunov function $V : \mathcal{X} \to \mathbb{R}_{\geq 0}$ as in Definition 4.1, then $\mathbf{x} = \mathbf{0}_n$ is an asymptotically stable equilibrium.*

See Appendix A for the proof (adapted from [Fürnsinn et al., 2023, Theorem 2]), and Appendix B for an explicit construction of a classical one-step Lyapunov function from a generalized Lyapunov function when the step weights $\{\sigma_i\}_{i=1}^M$ are state-independent.

**Remark 4.3** (**Relation to $k$-Inductive Verification**). *The generalized Lyapunov condition in (9) can be viewed through the lens of $k$-step inductive verification [Brain et al., 2015, Anand et al., 2022, Wooding and Lavaei, 2024]. In particular, by selecting weights $\sigma_1 = \cdots = \sigma_{k-1} = 0$ and $\sigma_k = M$, condition (9) reduces to $V(x_{t+k}) \leq (1-\alpha)V(x_t)$, which is analogous to a $k$-step Lyapunov decrease condition used in $k$-inductive reasoning. Classical $k$-induction requires a strict decrease at the final step while permitting bounded increases in intermediate steps. In contrast, Definition 4.1 enforces a decrease in a weighted average across multiple steps, with the weights $\sigma_i$ chosen adaptively or learned during training. This distinction provides additional flexibility, enabling certificates that may be easier to construct for nonlinear systems with learned controllers.*

Returning to the LQR setting in Section 3, we consider certifying the stability of (6) via a generalized Lyapunov function. Building on Theorem 3.2, we augment $J_\gamma^*$ with a quadratic residual and require the composite function to satisfy (9), leading to a new set of LMIs for stability certification.

**Theorem 4.4** (**Generalized Lyapunov Stability for Discounted LQR via LMIs**). *Suppose Assumption 3.1 holds. Consider an optimal policy $\boldsymbol{\pi}_\gamma^*$ and value function $J_\gamma^*$ obtained from the LQR problem with discount $\gamma$. Let $\mathbf{P}$ denote the solution to the undiscounted ARE in (5) (with $\gamma = 1$). If there exist symmetric positive definite matrices $\mathbf{S}_0, \mathbf{S}_1, \ldots, \mathbf{S}_M$, scalars $\varpi, \alpha_1, \ldots, \alpha_M > 0$, and weights $\sigma_1, \ldots, \sigma_M \geq 0$ satisfying $\sum_{i=1}^M \sigma_i \geq M$, such that:*

$$\begin{bmatrix} \frac{\sigma_1}{M}\mathbf{A}^\top \mathbf{S}_0 \mathbf{A} - \mathbf{S}_0 + \mathbf{S}_1 - \varpi\mathbf{Q} & \frac{\sigma_1}{M}\mathbf{A}^\top \mathbf{S}_0 \mathbf{B} \\ \frac{\sigma_1}{M}\mathbf{B}^\top \mathbf{S}_0 \mathbf{A} & \frac{\sigma_1}{M}\mathbf{B}^\top \mathbf{S}_0 \mathbf{B} - \varpi\mathbf{R} \end{bmatrix} \preceq 0, \tag{10a}$$

$$\begin{bmatrix} \frac{\sigma_{i+1}}{M}\mathbf{A}^\top \mathbf{S}_0 \mathbf{A} - \mathbf{S}_{i+1} - \varpi\mathbf{Q} & \frac{\sigma_{i+1}}{M}\mathbf{A}^\top \mathbf{S}_0 \mathbf{B} \\ \frac{\sigma_{i+1}}{M}\mathbf{B}^\top \mathbf{S}_0 \mathbf{A} & \frac{\sigma_{i+1}}{M}\mathbf{B}^\top \mathbf{S}_0 \mathbf{B} - \varpi\mathbf{R} \end{bmatrix} \preceq 0, \quad \forall i = 1, \ldots, M-1, \tag{10b}$$

$$\alpha_i \mathbf{P} \preceq \mathbf{S}_i, \quad \forall i = 1, \ldots, M, \tag{10c}$$

*then the function*

$$V(\mathbf{x}) := J_\gamma^*(\mathbf{x}) + \frac{1}{\varpi}\mathbf{x}^\top \mathbf{S}_0 \mathbf{x} \tag{11}$$

*is a generalized Lyapunov function in the sense of Definition 4.1 (with constant weights), and certifies that the origin is globally exponentially stable for the closed-loop system (6), provided that:*

$$\gamma > \max\left( \frac{\sigma_M \varpi}{M\alpha_M}, \ldots, \frac{\sigma_2 \varpi}{M\alpha_2}, \frac{\sigma_1 \varpi}{M(\varpi + \alpha_1)} \right). \tag{12}$$

The proof is provided in Appendix C.

**Remark 4.5** (**Feasibility of Multi-Step LMIs**). *Note that when $\sigma_1 = M$ and $\sigma_i = 0$ for all $2 \leq i \leq M$, Theorem 4.4 recovers the classical Lyapunov result in Theorem 3.2. Therefore, if the original one-step LMIs (7) are feasible, then feasible solutions for (10) also exist. Empirically, the multi-step formulation can certify stability in additional cases where the one-step condition is infeasible, effectively enlarging the set of stabilizable systems. Formal characterization of this weaker feasibility property is left for future work.*

**Remark 4.6** (**Choice of $\sigma_i$ Weights**). *While the multi-step formulation enables optimizing the weights $\sigma_1, \ldots, \sigma_M$ to minimize the certified lower bound on $\gamma$ in (12), finding globally optimal weights is challenging due to its non-convexity nature. In practice, heuristics such as grid search for small $M$ and random sampling with local refinements for larger $M$ are sufficient to achieve noticeable improvements over the one-step baseline.*

**Example 4.7** (**Example 3.3 Revisited**). *We illustrate the benefits of the generalized multi-step LMIs from Theorem 4.4 in Example 3.3. We first focus on the case $M = 2$. Figure 1 shows how the certified bound on $\gamma$ varies with the choice of weights $\sigma_1$ and $\sigma_2$, constrained by $\sigma_1 + \sigma_2 = 2$. Since the undiscounted value function ($\gamma = 1$) satisfies the Lyapunov condition, this bound is capped at $\gamma = 1$. The classical Lyapunov setting $(\sigma_1, \sigma_2) = (2, 0)$ yields a conservative bound of $\gamma > 0.8090$, whereas optimizing over $\sigma_1$ and $\sigma_2$ improves the bound to $\gamma > 0.6229$ at $(\sigma_1, \sigma_2) = (1.54, 0.46)$. Another important observation is that increasing $M$ significantly reduces the certified lower bound on $\gamma$, progressively approaching the true stability threshold $\gamma^\star = 1/3$, as shown in Figure 2.*

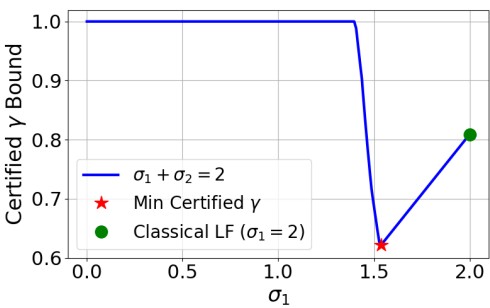

Figure 1: Certified $\gamma$ bound for $M = 2$.

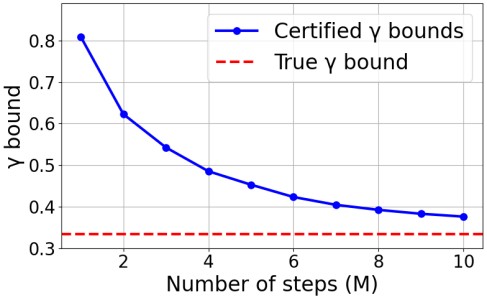

Figure 2: Certified $\gamma$ bound versus $M$.

## 5 Nonlinear Systems: Stability Certification of RL Policies

Inspired by the treatment of linear systems described in Section 4, we first formulate a post-hoc approach to certify stability of policies for nonlinear systems obtained by RL using a generalized Lyapunov function. The key observation from Theorem 4.4 is that augmenting the optimal value function with a residual term can result in a valid generalized Lyapunov function, as in (11). Here, we use the same idea to form generalized Lyapunov functions for nonlinear systems with unknown dynamics, which is a problem considered by RL. The value function and policy obtained by an RL algorithm are typically parameterized by neural networks. Let $\boldsymbol{\pi}_{\text{RL}}$ be a pre-trained RL policy (e.g., obtained by [Schulman et al., 2017, Haarnoja et al., 2018, Hansen et al., 2022]), and let $J_\gamma^{\boldsymbol{\pi}_{\text{RL}}}$ denote its corresponding learned value function. We consider a generalized Lyapunov candidate as:

$$V(\mathbf{x}; \boldsymbol{\theta}_1) = J_\gamma^{\boldsymbol{\pi}_{\text{RL}}}(\mathbf{x}) + \varphi(\mathbf{x}; \boldsymbol{\theta}_1), \tag{13}$$

where $\varphi(\mathbf{x}; \boldsymbol{\theta}_1)$ is a neural residual correction. To allow more flexibility in the generalized Lyapunov condition (9), we introduce a step-weighting network $\sigma(\mathbf{x}; \boldsymbol{\theta}_2) \in \mathbb{R}_{\geq 0}^M$ that outputs non-negative weights over a rollout horizon of length $M$. The weights are required to satisfy $\sum_{i=1}^M \sigma_i(\mathbf{x}; \boldsymbol{\theta}_2) \geq M$. These weights are then used to construct the generalized Lyapunov decrease condition:

$$F(\mathbf{x}_k) := \frac{1}{M}\left(\sum_{i=1}^M \sigma_i(\mathbf{x}_k; \boldsymbol{\theta}_2)\, V(\mathbf{x}_{k+i}; \boldsymbol{\theta}_1)\right) - (1 - \bar{\alpha})\, V(\mathbf{x}_k; \boldsymbol{\theta}_1), \tag{14}$$

where $\bar{\alpha} \in (0, 1)$ is a user-specified decay parameter. We train the networks $\varphi(\cdot; \boldsymbol{\theta}_1)$ and $\sigma(\cdot; \boldsymbol{\theta}_2)$ jointly by minimizing a loss that penalizes violations of the generalized Lyapunov condition:

$$\mathcal{L}(\boldsymbol{\theta}_1, \boldsymbol{\theta}_2) := \frac{1}{N}\sum_{i=1}^N \text{ReLU}\big(F(\mathbf{x}_k^{(i)})\big), \tag{15}$$

where $\{\mathbf{x}_k^{(i)}\}_{i=1}^N$ are sampled initial states.

**Remark 5.1** (**Network Architecture**). *In practice, the learned value function $J_\gamma^{\boldsymbol{\pi}_{RL}}$ may not be optimal, hence, may not satisfy $J_\gamma^{\boldsymbol{\pi}_{RL}}(\mathbf{0}_n) = 0$. To ensure that the generalized Lyapunov candidate satisfies $V(\mathbf{0}_n; \boldsymbol{\theta}_1) = 0$ and is positive definite, we modify (13) as:*

$$V(\mathbf{x}; \boldsymbol{\theta}_1) := \left| J_\gamma^{\boldsymbol{\pi}_{RL}}(\mathbf{x}) - J_\gamma^{\boldsymbol{\pi}_{RL}}(\mathbf{0}_n)\right| + \left|\varphi(\mathbf{x}; \boldsymbol{\theta}_1) - \varphi(\mathbf{0}_n; \boldsymbol{\theta}_1)\right| + \beta\|\mathbf{x}\|^2, \tag{16}$$

*where $\beta > 0$ is a small constant (the term $\beta\|\mathbf{x}\|^2$ enforces strict positive definiteness for $\mathbf{x} \neq \mathbf{0}_n$ and improves numerical stability near the origin.) The step-weighting network $\sigma(\mathbf{x}; \boldsymbol{\theta}_2) \in \mathbb{R}_{\geq 0}^M$ ends with a softmax layer scaled by $M$, ensuring the output weights satisfy $\frac{1}{M}\sum_{i=1}^M \sigma_i(\mathbf{x}; \boldsymbol{\theta}_2) = 1$.*

**Remark 5.2 (Equilibrium Behavior).** *In practice, RL policies often converge to a neighborhood near the origin rather than the origin itself. Thus, we train and verify the generalized Lyapunov condition over the set $\mathcal{X} \setminus \mathcal{B}(\mathbf{0}_n; \delta)$, where $\mathcal{B}(\mathbf{0}_n; \delta)$ denotes a ball of radius $\delta$ around the origin.*

**Training and Evaluation Setup.** We evaluate our method on two standard RL control benchmarks from Gymnasium [Towers et al., 2024] and the DeepMind Control Suite [Tassa et al., 2018]. RL policies $\boldsymbol{\pi}_{\mathrm{RL}}$ and their corresponding value functions $J_\gamma^{\boldsymbol{\pi}_{\mathrm{RL}}}$ are trained using implementations from Raffin et al. [2021], Hansen et al. [2022]. We collect a total of $N$ rollout trajectories by simulating the closed-loop system under $\boldsymbol{\pi}_{\mathrm{RL}}$, with each trajectory $\tau^{(i)} = \{\mathbf{x}_k^{(i)}\}_{k=0}^M$ starting from a randomly sampled initial state $\mathbf{x}_0^{(i)}$. Additional training and architecture details are provided in Appendix D.

**Inverted Pendulum Swingup.** We consider the inverted pendulum environment from Towers et al. [2024] with parameters $m = 1$, $l = 1$, $g = 10$, and control limits $|u| \leq \frac{mgl}{5} = 2$. The state space is $[-\pi, \pi) \times [-8, 8]$. Due to tight torque limits, the pendulum must swing back and forth to build momentum before reaching the upright position. This makes it challenging to synthesize a policy with a Lyapunov certificate valid over the entire state space. Prior work [Yang et al., 2024, Long et al., 2024] has shown that regions where stability can be verified are typically restricted to small neighborhoods near the upright position, failing to cover the full swing-up trajectories. In contrast, modern RL policies can discover effective swing-up behaviors but lack stability guarantees. Our method bridges this gap: we take a trained SAC policy Haarnoja et al. [2018] and apply our generalized Lyapunov function training with $M = 15$. Figure 3 shows the generalized Lyapunov function values along several trajectories from different initial states. As expected, the function exhibits non-monotonic behavior but shows an overall decline over the planning horizon. Figure 4 visualizes the Lyapunov function across the state space. Figure 5 plots the residual $F(\mathbf{x}_k)$ defined in (14), verifying that the generalized decrease condition is satisfied throughout the domain.

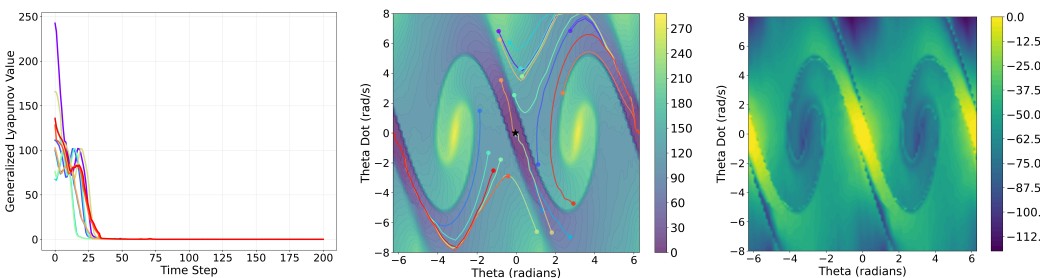

Figure 3: Generalized Lyapunov function values along trajectories.

Figure 4: Generalized Lyapunov function value over the state space.

Figure 5: Generalized Lyapunov decrease condition.

**Cartpole Swingup.** We consider the cartpole swing-up task from Tassa et al. [2018]. The system state is represented as $[x, \cos(\theta), \sin(\theta), \dot{x}, \dot{\theta}]$, where $x$ is the cart position, $\theta$ is the pole angle, and the remaining terms are their velocities. The goal is to swing the pole upright $\theta = 0$ and stabilize the position at $x = 0$. We use a TD-MPC policy [Hansen et al., 2022] and apply our certificate training (15) with $M = 20$. Figure 6 visualize the generalized Lyapunov condition across three representative 2D slices of the state space, with the remaining two states fixed at zero. The generalized decrease condition is satisfied throughout these slices, suggesting asymptotic stability of the RL policy.

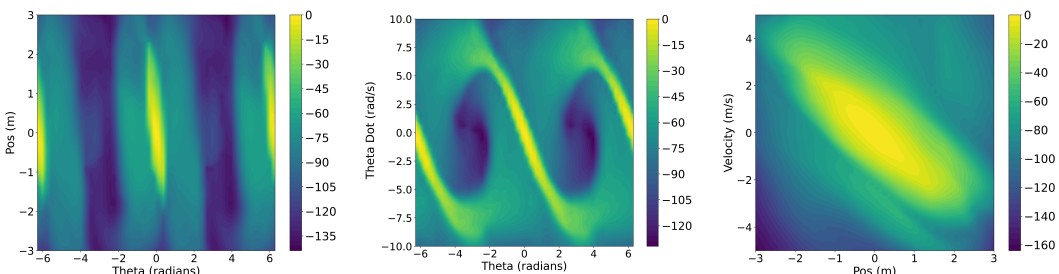

Figure 6: Generalized Lyapunov decrease condition for the cartpole swing-up using a TD-MPC policy.

In addition to qualitative visualizations, we quantitatively evaluate our learned certificates by sampling $N_{\text{test}}$ states from the full state space and checking whether (14) is satisfied. For each environment, we train certificates for multiple RL policies (the PPO policy fails to stabilize the cartpole from some initial states). As shown in Table 1, the condition holds for all test states in both environments.

Table 1: Percentage of sampled test states satisfying $F(\mathbf{x}_k) \leq 0$ under different RL policies.

| Environment | RL Methods | $M$ | $N_{\text{test}}$ | % Satisfying $F(\mathbf{x}_k) \leq 0$ |
|---|---|---|---|---|
| Inverted Pendulum | PPO, SAC, TD-MPC | 15 | 10,000 | 100% |
| Cartpole | SAC, TD-MPC | 20 | 1,000,000 | 100% |

**Concentration of Learned Step-Weights.** To further analyze the learned generalized Lyapunov function, we examine the distribution of the learned step-weights across the horizon. For each trained policy, the step-weight network was evaluated over $10,000$ uniformly sampled states from the state space, and the normalized weights were averaged across samples. The weights were then grouped into five equal bins corresponding to different segments of the Lyapunov horizon. Table 2 summarizes the fraction of the total weight assigned to each bin.

Table 2: Average fraction of total step-weight assigned to each portion of the rollout horizon. Results are averaged over 10,000 test states for each policy.

| Environment + Policy | Steps 0–20% | 20–40% | 40–60% | 60–80% | 80–100% |
|---|---|---|---|---|---|
| Pendulum + PPO | 12.3 | 11.4 | 16.7 | 21.8 | **37.8** |
| Pendulum + SAC | 10.2 | 13.3 | 17.5 | 22.6 | **36.4** |
| Cartpole + SAC | 10.8 | 10.5 | 16.2 | 25.6 | **36.1** |
| Cartpole + TD-MPC | 14.1 | 13.9 | 16.8 | 24.7 | **30.5** |

As shown in Table 2, the consistent concentration of weights toward the end of the horizon reveals a key strategy learned by our method. By assigning lower weights to initial steps and higher weights to later ones, the network effectively learns to tolerate initial non-monotonic transients in the Lyapunov candidate, focusing instead on ensuring a decisive, weighted decrease over the full horizon. This behavior empirically realizes the central benefit of the generalized Lyapunov condition: providing a flexible yet formal basis for certifying complex neural policies.

## 6 Joint Synthesis of Stable Neural Policies and Certificates

So far, we addressed the certification of stability for fixed pre-trained control policies. In this section, we consider *joint synthesis* of neural controllers and Lyapunov certificates and employ formal verification inspired by Chang et al. [2019], Dai et al. [2021], Wu et al. [2023], Yang et al. [2024]. Compared to prior methods, our generalized Lyapunov framework naturally extends this setting by replacing the standard one-step decrease condition with the proposed multi-step, weighted formulation in (9).

**Definition 6.1** (**Region of Attraction**). The *region of attraction (ROA)* $\mathcal{R} \subseteq \mathbb{R}^n$ of the (locally asymptotically stable) origin for the discrete-time system (3) is the set of all points from which the system trajectory converges to the origin, i.e., $\mathbf{x}_0 \in \mathcal{R}$ implies $\lim_{k \to \infty} \mathbf{x}_k = \mathbf{0}_n$.

In general, precise characterizations of $\mathcal{R}$ are challenging to obtain and one instead looks for suitable approximations. Here, we seek to jointly learn a policy $\boldsymbol{\pi}(\mathbf{x}; \boldsymbol{\phi})$ and a certificate function $V(\mathbf{x}; \boldsymbol{\theta}_1)$ such that the closed-loop system $\mathbf{x}_{k+1} = \mathbf{f}(\mathbf{x}_k, \boldsymbol{\pi}_\phi(\mathbf{x}_k))$ is provably asymptotically stable and obtain at the same time an inner approximation $\mathcal{S}$ of the ROA $\mathcal{R}$. Yang et al. [2024] formalized it as a constrained optimization that maximizes the volume of a Lyapunov sublevel set $\mathcal{S} = \{\mathbf{x} \in \mathbb{R}^n : V(\mathbf{x}) \leq \rho\}$ under Lyapunov constraints

$$\max_{\boldsymbol{\theta}_1, \boldsymbol{\phi}} \quad \text{Vol}(\mathcal{S}) \tag{17a}$$

$$\text{s.t.} \quad V(\mathbf{0}_n; \boldsymbol{\theta}_1) = 0, \quad V(\mathbf{x}; \boldsymbol{\theta}_1) > 0 \quad \forall \mathbf{x} \in \mathcal{S} \setminus \{\mathbf{0}_n\}, \tag{17b}$$

$$V(\mathbf{x}_{k+1}; \boldsymbol{\theta}_1) - (1 - \bar{\alpha}) V(\mathbf{x}_k; \boldsymbol{\theta}_1) \leq 0 \quad \forall \mathbf{x}_k \in \mathcal{S}, \tag{17c}$$

where $\bar{\alpha} \in (0, 1)$ is a user-specified decay parameter. To utilize a generalized Lyapunov function, we replace (17c) with the generalized $M$-step condition for each $\mathbf{x}_k \in \mathcal{S}$, with $\sum_{i=1}^{M} \sigma_i(\mathbf{x}_k) \geq M$:

$$F(\mathbf{x}_k) := \frac{1}{M}\left(\sum_{i=1}^{M} \sigma_i(\mathbf{x}_k)V(\mathbf{x}_{k+i}; \boldsymbol{\theta}_1)\right) - (1 - \bar{\alpha})V(\mathbf{x}_k; \boldsymbol{\theta}_1) \leq 0, \quad \sigma_i(\mathbf{x}_k) \geq \underline{\sigma} > 0, \quad (18)$$

where $\underline{\sigma} \in \mathbb{R}_{>0}$ is a uniform lower bound on the weights. The condition (18) enables learning certificates that tolerate non-monotonic behavior along a trajectory, but it no longer guarantees that $\mathcal{S}$ is *forward invariant*. However, as Theorem 6.2 shows, $\mathcal{S}$ still defines a valid inner approximation of the ROA and guarantees asymptotic stability of the origin.

**Theorem 6.2** (**Asymptotic Stability Relative to** $\mathcal{S}$)**.** *If the optimization problem* (17) *is solved with the generalized condition* (18) *instead of* (17c)*, then* $\mathcal{S} \subset \mathcal{R}$*.*

See Appendix E for the proof.

**Training Formulation.** Following [Yang et al., 2024], we reformulate the problem (17) into a learning objective by sampling states $\mathbf{x}_k \in \mathcal{X}$ and penalizing violations of the Lyapunov conditions using soft constraints. For each $\mathbf{x}_k \in \mathcal{D}$, we simulate $M$ forward steps under the closed-loop dynamics and compute the generalized Lyapunov residual $F(\mathbf{x}_k)$ as defined in (18).

To enforce stability and domain constraints, we define the stability loss:

$$\mathcal{L}_{\text{stab}}(\mathbf{x}_k) := \text{ReLU}\left(\min\left\{\text{ReLU}(F(\mathbf{x}_k)) + c_0 \mathcal{H}(\mathbf{x}_k), \rho - V(\mathbf{x}_k, \boldsymbol{\theta}_1)\right\}\right), \quad (19)$$

where $\mathcal{H}(\mathbf{x}_k) := \sum_{i=1}^{M} \|\text{ReLU}(\mathbf{x}_{k+i} - \mathbf{x}_{\text{up}}) + \text{ReLU}(\mathbf{x}_{\text{lo}} - \mathbf{x}_{k+i})\|_1$ penalizes violations of the bounded domain $\mathcal{X} = \{\mathbf{x} \mid \mathbf{x}_{\text{lo}} \leq \mathbf{x} \leq \mathbf{x}_{\text{up}}\}$. The inner $\min$ ensures that the generalized Lyapunov decrease condition is only enforced inside the certified region $\mathcal{S}$.

To expand the certifiable region, Yang et al. [2024] proposed a surrogate region loss $\mathcal{L}_{\text{region}} := \sum_{j=1}^{N} \text{ReLU}\left(V(\mathbf{x}_j; \boldsymbol{\theta}_1)/\rho - 1\right)$, where the candidate states $\mathbf{x}_j$ are obtained via random boundary sampling or projected gradient descent (PGD) to minimize $V(\cdot; \boldsymbol{\theta}_1)$. PGD is also used for falsification by maximizing stability violations, generating additional training states for $\mathcal{D}$.

The final training objective is $\mathcal{L}(\boldsymbol{\theta}_1, \boldsymbol{\phi}) := \sum_{\mathbf{x}_k \in \mathcal{D}} \mathcal{L}_{\text{stab}}(\mathbf{x}_k) + c_1 \mathcal{L}_{\text{region}} + c_2 \|\boldsymbol{\theta}_1, \boldsymbol{\phi}\|_1$, where $\mathcal{D}$ contains states sampled randomly and generated by falsification.

**Remark 6.3.** *Although the theoretical framework (Theorem 4.2) supports trainable weights $\sigma_i(\mathbf{x}_k)$, including neural network parameterizations (see Section 5), certifying stability under jointly learned controllers, Lyapunov functions, and weights remains computationally challenging with existing tools [Wang et al., 2021, Gao et al., 2013]. In practice, we fix $\sigma_i$ during training. For small values of $M$, we select the best-performing weight configuration via a simple grid search.*

**Verification Formulation.** After training, we verify that the generalized decrease condition holds within $\mathcal{X}$ using the $\alpha$-$\beta$-CROWN verifier [Wang et al., 2021]. We formally certify Lyapunov stability of the origin if the following holds for all $\mathbf{x}_k \in \mathcal{X}$:

$$\left(-F(\mathbf{x}_k) \geq 0 \ \wedge \ \bigwedge_{i=1}^{M} \mathbf{x}_{k+i} \in \mathcal{X}\right) \ \vee \ (V(\mathbf{x}_k) \geq \rho). \quad (20)$$

As Theorem 6.2 implies, if (20) holds for all $\mathbf{x}_k \in \mathcal{X}$, then $\mathcal{S}$ is an inner approximation of $\mathcal{R}$.

**Evaluation.** We evaluate our generalized Lyapunov synthesis approach on three systems presented in [Yang et al., 2024]: inverted pendulum, path tracking, and 2D quadrotor (with a 6D state space). Detailed system dynamics and neural network architectures used are provided in Appendix F.

Figures 7 and 8 show the certified stability regions $\mathcal{S}$ for different horizon lengths $M$. We use fixed step weights $(\sigma_1, \sigma_2, \dots)$ selected via grid search: for the inverted pendulum, $(0.4, 1.6)$ for $M{=}2$ and $(0.3, 1.5, 1.2)$ for $M{=}3$; for path tracking, $(0.4, 1.6)$ and $(1.2, 1.2, 0.6)$, respectively. As shown in the figures, multi-step training and verification yields consistently larger certifiable ROAs.

Table 3 reports the quantitative results, including the volume $(\mathcal{S})$ and the verification time. The volume is estimated via Monte Carlo integration: we sample $n = 10^6$ points in $\mathcal{X}$ and compute the fraction satisfying $V(\mathbf{x}) \leq \rho$, averaged across trials and scaled by the total domain volume. To assess

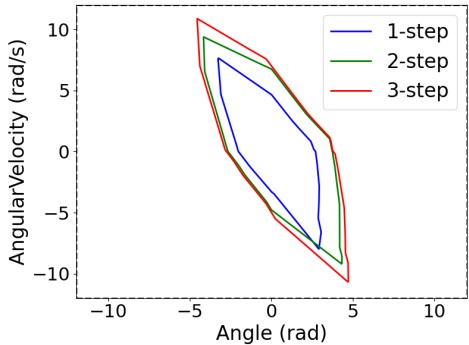

Figure 7: Certified ROAs for inverted pendulum.

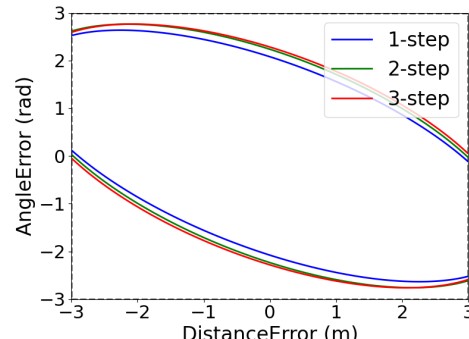

Figure 8: Certified ROAs for path-tracking.

robustness, we repeat each experiment 10 times with different random seeds and report the mean and standard deviation. The results confirm that our generalized stability formulation consistently certifies larger ROA volumes compared to the classical one-step Lyapunov approach. This improvement comes at the cost of longer verification times, since bounding multiple intermediate states and their weighted combinations increases the complexity of bound propagation and branching decisions.

Table 3: Certified region volume (left) and verification time (right) under different $M$-step Lyapunov training. Abbreviations: I.P. = Inverted Pendulum, P.T. = Path Tracking, 2.Q. = 2D Quadrotor.

| System | $M = 1$ | $M = 2$ | $M = 3$ | $M = 1$ | $M = 2$ | $M = 3$ |
|---|---|---|---|---|---|---|
| I.P. | $42.9 \pm 1.2$ | $76.7 \pm 1.3$ | $89.2 \pm 1.2$ | $11.7 \pm 0.5$ | $21.5 \pm 0.8$ | $39.2 \pm 1.3$ |
| P.T. | $21.8 \pm 0.6$ | $23.6 \pm 0.5$ | $23.9 \pm 0.5$ | $8.6 \pm 0.5$ | $19.5 \pm 1.0$ | $36.7 \pm 1.3$ |
| 2.Q. | $103.5 \pm 1.8$ | $109.1 \pm 2.0$ | $113.7 \pm 2.0$ | $2209.5 \pm 72.6$ | $3858.7 \pm 112.4$ | $5628.6 \pm 185.9$ |

(a) Certified ROA volume ($\mathcal{S}$).   (b) Verification time (seconds).

## 7    Conclusion, Limitations, and Future Work

We presented a new approach for constructing stability certificates for closed-loop systems under policies obtained from optimal control and RL. We also showed that our method can be used for joint synthesis of neural policies and stability certificates. Rather than learning certificates from scratch, a key insight is that the value function of a considered policy can be augmented with a residual term to obtain a certificate candidate. We also replaced the classical Lyapunov decrease condition with a generalized multi-step Lyapunov formulation, which provides asymptotical stability guarantees while improving learning flexibility. Our theoretical and empirical results show that this perspective enables construction of more effective stability certificates compared to traditional Lyapunov approaches.

Our approach has several limitations. First, the certification horizon $M$ is fixed during training, and it remains unclear how to determine the minimal viable $M$ for a given system. Second, the method has not yet been applied to high-dimensional systems such as humanoids or dexterous manipulators. Third, formally verifying neural policies (e.g., from RL), along with learned state-dependent weights and certificate functions, remains computationally challenging with existing tools.

This work opens up several promising directions for future research. One key avenue is to develop partial-state stability certification, providing guarantees for task-relevant components such as torso height or velocity in humanoid locomotion rather than for the full system state, which can improve both scalability and interpretability. A second direction is to extend the generalized Lyapunov formulation to certify robust stability under bounded perturbations and stochastic disturbances, taking inspiration from input-to-state and robust Lyapunov theory. Another important direction is to relax the requirement of knowing the equilibrium state, allowing stability analysis when the equilibrium is unknown or non-isolated, as often occurs in reinforcement learning, by inferring equilibria from data or from the reward structure. Finally, investigating the connection between generalized Lyapunov conditions and optimality objectives may yield new insights for reward design and policy refinement in reinforcement learning.

## Acknowledgments

We gratefully acknowledge support from ONR Award N00014-23-1-2353 and NSF Award CCF-2112665 (TILOS).

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

# A  Proof of Theorem 4.2

*Proof. Attractivity.* We first show attractivity. Fix any initial condition $\mathbf{x}_0 \in \mathcal{X} \setminus \{\mathbf{0}\}$ and let $(\mathbf{x}_k)_{k \geq 0}$ be the corresponding trajectory. For each $k \geq 0$ choose an index

$$\ell(k) \in \{1, \dots, M\} \quad \text{such that} \quad V\big(\mathbf{x}_{k+\ell(k)}\big) < V(\mathbf{x}_k), \tag{21}$$

whose existence is guaranteed by (9). Define the strictly increasing sequence of *return indices*

$$k_0 := 0, \qquad k_{j+1} := k_j + \ell\big(k_j\big), \quad j = 0, 1, \dots, \tag{22}$$

for which, by construction,

$$0 < k_{j+1} - k_j \leq M, \qquad V\big(\mathbf{x}_{k_{j+1}}\big) < V\big(\mathbf{x}_{k_j}\big), \quad j = 0, 1, \dots. \tag{23}$$

Due to the positive-definiteness of $V$, applying standard Lyapunov arguments on $\{V(x_{k_j})\}$, we conclude that $\lim_{k_j \to \infty} V(\mathbf{x}_{k_j}) = 0$.

Because the gaps satisfy $k_{j+1} - k_j \leq M$ and $V$ is continuous, we have

$$\lim_{k \to \infty} V(\mathbf{x}_k) = 0. \tag{24}$$

Finally, the positive-definiteness of $V$ yields $\mathbf{x}_k \to \mathbf{0}_n$ as $k \to \infty$. Thus the origin is *attractive*.

*Stability.* Since both the system dynamics $\mathbf{f}$ and the policy $\boldsymbol{\pi}$ are assumed to be Lipschitz, the closed-loop map $\mathbf{x} \mapsto \mathbf{f}(\mathbf{x}, \boldsymbol{\pi}(\mathbf{x}))$ is also Lipschitz on the forward-invariant set $\mathcal{X}$. Let $L > 0$ denote the Lipschitz constant. Then, for all $0 \leq i < M$ and $k \geq 0$,

$$\|\mathbf{x}_{k+i}\| \leq L^i \|\mathbf{x}_k\|. \tag{25}$$

Fix an arbitrary $\varepsilon > 0$ and choose

$$\delta := \frac{\varepsilon}{L^{M-1}}. \tag{26}$$

Consider any initial state with $\|\mathbf{x}_0\| < \delta$. By construction of the return indices $k_j$ in (22) we have $\|\mathbf{x}_{k_{j+1}}\| \leq \|\mathbf{x}_{k_j}\|$ for all $j$, so in particular

$$\|\mathbf{x}_{k_j}\| < \delta, \qquad j = 0, 1, \dots. \tag{27}$$

For an arbitrary time index $k$ pick $j$ such that $k_j \leq k < k_j + M$. Applying (25) with $i = k - k_j < M$ and (27) yields

$$\|\mathbf{x}_k\| \leq L^{M-1} \|\mathbf{x}_{k_j}\| < L^{M-1} \delta = \varepsilon.$$

Because $\varepsilon$ was arbitrary, the $\varepsilon$–$\delta$ condition holds and the origin is Lyapunov stable.

In conclusion, the origin is both attractive and Lyapunov–stable on $\mathcal{X}$; therefore it is asymptotically stable relative to $\mathcal{X}$. $\qquad \square$

**Remark A.1** (**Exponential Stability**). *Suppose the function $V : \mathcal{X} \to \mathbb{R}_{\geq 0}$ satisfies the positive definiteness condition in (4a), and there exist constants $\beta_1, \beta_2, p > 0$ such that*

$$\beta_1 \|\mathbf{x}\|^p \leq V(\mathbf{x}) \leq \beta_2 \|\mathbf{x}\|^p, \quad \forall \mathbf{x} \in \mathcal{X}. \tag{28}$$

*Suppose further that $V$ satisfies the strict generalized decrease condition for all $\mathbf{x}_k \in \mathcal{X}$,*

$$\frac{1}{M} \sum_{i=1}^{M} \sigma_i(\mathbf{x}_k) V(\mathbf{x}_{k+i}) - V(\mathbf{x}_k) \leq -\alpha(V(\mathbf{x}_k)), \tag{29}$$

*where $\alpha : \mathbb{R}_{\geq 0} \to \mathbb{R}_{\geq 0}$ is a class-$\mathcal{K}$ function. Then the origin $\mathbf{x} = \mathbf{0}_n$ is exponentially stable.*

# B Construction of a Classical Lyapunov Function from a Generalized One

This appendix shows that when the generalized $M$-step Lyapunov decrease in Definition 4.1 holds with *state-independent* step weights, one can explicitly construct a classical one-step Lyapunov function, following a similar idea to classical constructions that convert multi-step (non-monotone) decrease conditions into a one-step Lyapunov decrease **?**. In contrast, when the step weights depend on the state, obtaining an explicit closed-form classical Lyapunov function from the generalized certificate appears non-trivial; while asymptotic stability still follows from Theorem 4.2, deriving a simple one-step Lyapunov function construction in the state-dependent case is left for future work.

**Assumption B.1** (**Constant step weights**). *The step weights in Definition 4.1 are constant, i.e., there exist $\sigma_1, \ldots, \sigma_M \geq 0$ (independent of $\mathbf{x}$) such that for all $\mathbf{x} \in \mathcal{X} \setminus \{\mathbf{0}\}$,*

$$\frac{1}{M}\Big(\sum_{i=1}^{M} \sigma_i V(\mathbf{x}_i)\Big) - V(\mathbf{x}) < 0, \quad \frac{1}{M}\sum_{i=1}^{M} \sigma_i \geq 1, \tag{30}$$

*where $\mathbf{x}_0 = \mathbf{x}$ and $\mathbf{x}_{i+1} = \mathbf{f}(\mathbf{x}_i, \boldsymbol{\pi}(\mathbf{x}_i))$ for $i = 0, \ldots, M-1$.*

**Lemma B.2** (**Explicit construction of a classical Lyapunov function**). *Suppose $V : \mathcal{X} \to \mathbb{R}_{\geq 0}$ satisfies Definition 4.1 and Assumption B.1. Without loss of generality, assume*

$$\sum_{i=1}^{M} \sigma_i = M. \tag{31}$$

*Define coefficients*

$$a_j := \frac{1}{M}\sum_{i=j+1}^{M} \sigma_i, \qquad j = 0, \ldots, M-1, \tag{32}$$

*and the function*

$$W(\mathbf{x}) := \sum_{j=0}^{M-1} a_j V\big(\mathbf{f}^j(\mathbf{x})\big), \tag{33}$$

*where $\mathbf{f}^j$ denotes $j$-fold composition of the closed-loop map $\mathbf{x} \mapsto \mathbf{f}(\mathbf{x}, \boldsymbol{\pi}(\mathbf{x}))$. Then $W$ is a classical Lyapunov function:*

$$W(\mathbf{0}) = 0, \qquad W(\mathbf{x}) > 0 \ \forall \mathbf{x} \neq \mathbf{0}, \tag{34}$$

*and it decreases strictly in one step:*

$$W(\mathbf{f}(\mathbf{x}, \boldsymbol{\pi}(\mathbf{x}))) - W(\mathbf{x}) < 0 \ \forall \mathbf{x} \neq \mathbf{0}. \tag{35}$$

*Proof.* By (31) and (32), we have $a_0 = 1$ and $a_j \geq 0$ for all $j$. Hence

$$W(\mathbf{x}) \geq V(\mathbf{x}), \tag{36}$$

which implies (34). If $V$ is radially unbounded, then so is $W$ by (36).

Let $\{\mathbf{x}_k\}_{k\geq 0}$ be any closed-loop trajectory and define $V_k := V(\mathbf{x}_k)$. Using (33),

$$\begin{aligned}
W(\mathbf{x}_{k+1}) - W(\mathbf{x}_k) &= \sum_{j=0}^{M-1} a_j V_{k+1+j} - \sum_{j=0}^{M-1} a_j V_{k+j} \\
&= -a_0 V_k + \sum_{j=1}^{M-1} (a_{j-1} - a_j) V_{k+j} + a_{M-1} V_{k+M}.
\end{aligned} \tag{37}$$

By (32), $a_{j-1} - a_j = \sigma_j / M$ for $j = 1, \ldots, M-1$ and $a_{M-1} = \sigma_M / M$, and by (31) we have $a_0 = 1$. Substituting into (37) gives

$$W(\mathbf{x}_{k+1}) - W(\mathbf{x}_k) = \frac{1}{M}\sum_{i=1}^{M} \sigma_i V_{k+i} - V_k. \tag{38}$$

The right-hand side of (38) is strictly negative for all $\mathbf{x}_k \neq \mathbf{0}$ by (30), proving (35). $\qquad \square$

**Scaling the weights.** If $\sum_{i=1}^{M} \sigma_i > M$, define $\tilde{\sigma}_i := \frac{M}{\sum_{j=1}^{M} \sigma_j} \sigma_i$. Then $\tilde{\sigma}_i \geq 0$ and $\sum_{i=1}^{M} \tilde{\sigma}_i = M$. Since $\frac{M}{\sum_{j=1}^{M} \sigma_j} \in (0,1)$, the left-hand side of (30) is scaled down while $V(\mathbf{x})$ is unchanged, so the strict inequality in (30) remains valid. Hence (31) is without loss of generality.

## C Proof of Theorem 4.4

For clarity, we present the detailed proof for the case $M = 2$. The general $M$-step case follows analogously by extending the argument to multiple future steps.

*Proof.* We consider the generalized Lyapunov decrease condition over two steps:

$$\frac{1}{2} \left( \sigma_2 V(\mathbf{x}_{k+2}) + \sigma_1 V(\mathbf{x}_{k+1}) \right) - V(\mathbf{x}_k) < 0, \tag{39}$$

where $\sigma_1, \sigma_2 \geq 0$ and $\sigma_1 + \sigma_2 \geq 2$.

Using the discounted Bellman equation for the optimal value function $V_\gamma$,

$$V_\gamma(\mathbf{x}_{k+1}) - V_\gamma(\mathbf{x}_k) = -\gamma^{-1} \ell(\mathbf{x}_k, \mathbf{u}_k^*) + (1-\gamma)\gamma^{-1} V_\gamma(\mathbf{x}_k), \tag{40}$$

we can recursively expand the two-step generalized decrease condition for $V_\gamma$. Direct computation yields:

$$\frac{1}{2} \left( \sigma_2 V_\gamma(\mathbf{x}_{k+2}) + \sigma_1 V_\gamma(\mathbf{x}_{k+1}) \right) - V_\gamma(\mathbf{x}_k)$$
$$= \frac{\sigma_2}{2} \gamma^{-1} V_\gamma(\mathbf{x}_{k+1}) + \left( \frac{\sigma_1}{2} \gamma^{-1} - 1 \right) V_\gamma(\mathbf{x}_k) - \frac{\sigma_1}{2} \gamma^{-1} \ell(\mathbf{x}_k, \mathbf{u}_k^*) - \frac{\sigma_2}{2} \gamma^{-1} \ell(\mathbf{x}_{k+1}, \mathbf{u}_{k+1}^*)$$
$$\leq \begin{bmatrix} \mathbf{x}_{k+1} & \mathbf{x}_k \end{bmatrix} \begin{bmatrix} \frac{\sigma_2}{2} \gamma^{-1} \mathbf{P} & \mathbf{0} \\ \mathbf{0} & (\frac{\sigma_1}{2}\gamma^{-1} - 1)\mathbf{P} \end{bmatrix} \begin{bmatrix} \mathbf{x}_{k+1} \\ \mathbf{x}_k \end{bmatrix} - \frac{\sigma_2}{2} \ell(\mathbf{x}_{k+1}, \mathbf{u}_{k+1}^*) - \frac{\sigma_1}{2} \ell(\mathbf{x}_k, \mathbf{u}_k^*) \quad (41)$$

However, due to the presence of the discount factor $\gamma \in (0,1)$, the value function $V_\gamma$ does not naturally satisfy the required decrease condition.

Following the approach in [Postoyan et al., 2017], we introduce an auxiliary quadratic function:

$$W(\mathbf{x}) = \frac{1}{\varpi} \mathbf{x}^\top \mathbf{S}_0 \mathbf{x}, \tag{42}$$

where $\mathbf{S}_0 \succ 0$ and $\varpi > 0$ are design parameters.

Define the composite function:

$$Y_\gamma(\mathbf{x}) := V_\gamma(\mathbf{x}) + W(\mathbf{x}). \tag{43}$$

Our goal is to ensure that $Y_\gamma$ satisfies the generalized Lyapunov decrease condition in (39).

We compute the change in the auxiliary function $W(\mathbf{x})$ over two steps:

$$\frac{1}{2} \left( \sigma_2 W(\mathbf{x}_{k+2}) + \sigma_1 W(\mathbf{x}_{k+1}) \right) - W(\mathbf{x}_k)$$
$$= \frac{1}{\varpi} \left( \frac{\sigma_2}{2} \mathbf{x}_{k+2}^\top \mathbf{S}_0 \mathbf{x}_{k+2} + \frac{\sigma_1}{2} \mathbf{x}_{k+1}^\top \mathbf{S}_0 \mathbf{x}_{k+1} - \mathbf{x}_k^\top \mathbf{S}_0 \mathbf{x}_k \right). \tag{44}$$

Expanding the system dynamics

$$\mathbf{x}_{k+1} = \mathbf{A}\mathbf{x}_k + \mathbf{B}\mathbf{u}_k, \quad \mathbf{x}_{k+2} = \mathbf{A}\mathbf{x}_{k+1} + \mathbf{B}\mathbf{u}_{k+1}, \tag{45}$$

and substituting into (44), we express the change in $W$ as

$$\frac{1}{2} \left( \sigma_2 W(\mathbf{x}_{k+2}) + \sigma_1 W(\mathbf{x}_{k+1}) \right) - W(\mathbf{x}_k)$$
$$= \frac{\sigma_1}{2\varpi} \left( \mathbf{x}_k^\top (\mathbf{A}^\top \mathbf{S}_0 \mathbf{A} - \mathbf{S}_0)\mathbf{x}_k + \mathbf{u}_k^\top \mathbf{B}^\top \mathbf{S}_0 \mathbf{B} \mathbf{u}_k + 2\mathbf{x}_k^\top \mathbf{A}^\top \mathbf{S}_0 \mathbf{B} \mathbf{u}_k \right)$$
$$+ \frac{\sigma_2}{2\varpi} \left( \mathbf{x}_{k+1}^\top \mathbf{A}^\top \mathbf{S}_0 \mathbf{A} \mathbf{x}_{k+1} + \mathbf{u}_{k+1}^\top \mathbf{B}^\top \mathbf{S}_0 \mathbf{B} \mathbf{u}_{k+1} + 2\mathbf{x}_{k+1}^\top \mathbf{A}^\top \mathbf{S}_0 \mathbf{B} \mathbf{u}_{k+1} \right). \tag{46}$$

To upper bound this change, we introduce positive definite matrices $\mathbf{S}_1, \mathbf{S}_2 \succ 0$ and impose:

$$\frac{1}{2}\left(\sigma_2 W(\mathbf{x}_{k+2}) + \sigma_1 W(\mathbf{x}_{k+1})\right) - W(\mathbf{x}_k)$$
$$\leq -\frac{1}{\varpi}\mathbf{x}_{k+1}^\top \mathbf{S}_2 \mathbf{x}_{k+1} - \frac{1}{\varpi}\mathbf{x}_k^\top \mathbf{S}_1 \mathbf{x}_k + \frac{\sigma_1}{2}\ell(\mathbf{x}_k, \mathbf{u}_k^*) + \frac{\sigma_2}{2}\ell(\mathbf{x}_{k+1}, \mathbf{u}_{k+1}^*). \tag{47}$$

Combining the changes in $V_\gamma$ and $W$, we obtain:

$$\frac{1}{2}\left(\sigma_2 Y_\gamma(\mathbf{x}_{k+2}) + \sigma_1 Y_\gamma(\mathbf{x}_{k+1})\right) - Y_\gamma(\mathbf{x}_k)$$
$$\leq -\mathbf{x}_{k+1}^\top \left(\frac{1}{\varpi}\mathbf{S}_2 - \frac{\sigma_2}{2}\gamma^{-1}\mathbf{P}\right)\mathbf{x}_{k+1} - \mathbf{x}_k^\top \left(\frac{1}{\varpi}\mathbf{S}_1 - \left(\frac{\sigma_1}{2}\gamma^{-1} - 1\right)\mathbf{P}\right)\mathbf{x}_k. \tag{48}$$

Thus, to ensure strict decay of $Y_\gamma$, it suffices to require:

$$\frac{1}{\varpi}\mathbf{S}_2 - \frac{\sigma_2}{2}\gamma^{-1}\mathbf{P} \succ 0, \tag{49}$$

$$\frac{1}{\varpi}\mathbf{S}_1 - \left(\frac{\sigma_1}{2}\gamma^{-1} - 1\right)\mathbf{P} \succ 0. \tag{50}$$

This leads to the requirement that $\gamma$ must satisfy:

$$\gamma > \max\left(\frac{\sigma_2\varpi}{2\alpha_2}, \quad \frac{\sigma_1\varpi}{2(\varpi + \alpha_1)}\right), \tag{51}$$

where $\alpha_1, \alpha_2 > 0$ satisfy the comparison inequalities:

$$\alpha_1 \mathbf{P} \preceq \mathbf{S}_1, \quad \alpha_2 \mathbf{P} \preceq \mathbf{S}_2. \tag{52}$$

Furthermore, by comparing (46) and (47), the auxiliary matrices must satisfy the following LMIs:

$$\begin{bmatrix} \frac{\sigma_1}{2}\mathbf{A}^\top \mathbf{S}_0 \mathbf{A} - \mathbf{S}_0 + \mathbf{S}_1 - \varpi\mathbf{Q} & \frac{\sigma_1}{2}\mathbf{A}^\top \mathbf{S}_0 \mathbf{B} \\ \frac{\sigma_1}{2}\mathbf{B}^\top \mathbf{S}_0 \mathbf{A} & \frac{\sigma_1}{2}\mathbf{B}^\top \mathbf{S}_0 \mathbf{B} - \varpi\mathbf{R} \end{bmatrix} \preceq 0, \tag{53}$$

$$\begin{bmatrix} \frac{\sigma_2}{2}\mathbf{A}^\top \mathbf{S}_0 \mathbf{A} - \mathbf{S}_2 - \varpi\mathbf{Q} & \frac{\sigma_2}{2}\mathbf{A}^\top \mathbf{S}_0 \mathbf{B} \\ \frac{\sigma_2}{2}\mathbf{B}^\top \mathbf{S}_0 \mathbf{A} & \frac{\sigma_2}{2}\mathbf{B}^\top \mathbf{S}_0 \mathbf{B} - \varpi\mathbf{R} \end{bmatrix} \preceq 0. \tag{54}$$

This concludes the proof. $\qquad\square$

## D   RL Certificates Training Details

### D.1   Network Architectures

We use feedforward neural networks with LeakyReLU activations for both the residual network $\varphi(\cdot, \boldsymbol{\theta}_1)$ and step-weighting network $\sigma(\cdot, \boldsymbol{\theta}_2)$. The architectures are summarized in Table 4. The residual network is initialized using Kaiming initialization [He et al., 2015], and all biases are initialized to zero.

Table 4: Neural network architectures for the residual and step-weighting networks.

| Component | Architecture | Output |
|---|---|---|
| Residual Network | 3 layers, width 64 | Scalar value |
| Step-weighting Network | 3 layers, width 64 | $M$-dimensional weights |

### D.2   Training Configuration

We train all networks using the Adam optimizer with an initial learning rate of $5 \times 10^{-4}$, a ReduceLROnPlateau scheduler (factor 0.5, patience 500), and a batch size of 256 for 1000 epochs. Gradients are clipped at 5.0. We set the decay parameter to $\bar{\alpha} = 0.02$ and the slack to $\beta = 0.01$. As discussed in Remark 5.2, we exclude a small ball around the origin during both training and evaluation. Specifically, we set $\delta = 0.05$ for the inverted pendulum and $\delta = 0.5$ for the cartpole.

### D.3 Compute Resources

All experiments were run on a single workstation with an NVIDIA RTX 4090 GPU, AMD Ryzen 9 7950X CPU, and 64 GB RAM. Training the Lyapunov certificate for the inverted pendulum takes approximately 40 minutes, and for the cartpole environment around 4 hours. These durations include all rollouts and training steps for both the residual and step-weight networks.

## E  Proof of Theorem 6.2

For clarity, we prove the result for $M = 2$ and assume $\sigma_1, \sigma_2$ are constants.

*Proof.* We restate the assumptions. Let the closed-loop system be:

$$\mathbf{x}_{k+1} = \mathbf{f}(\mathbf{x}_k, \boldsymbol{\pi}_\phi(\mathbf{x}_k)),$$

and $V : \mathbb{R}^n \to \mathbb{R}_{\geq 0}$ is a continuous function satisfying

$$V(\mathbf{0}_n) = 0, \quad V(\mathbf{x}_k) > 0 \quad \forall \mathbf{x}_k \in \mathcal{S} \setminus \{\mathbf{0}_n\}, \tag{55}$$

$$\frac{1}{2}(\sigma_1 V(\mathbf{x}_{k+1}) + \sigma_2 V(\mathbf{x}_{k+2})) \leq (1 - \bar{\alpha})V(\mathbf{x}_k), \quad \forall \mathbf{x}_k \in \mathcal{S}, \tag{56}$$

with $\sigma_1, \sigma_2 > \underline{\sigma} > 0$ and $\sigma_1 + \sigma_2 \geq 2$.

*Attractivity.* The attractivity proof follows as in Appendix A, we conclude $\lim_{k \to \infty} \mathbf{x}_k = \mathbf{0}_n$.

*Stability.* We follow the classical $\varepsilon$–$\delta$ Lyapunov proof in [Khalil, 1996, Thm. 3.3]. Fix an arbitrary radius $\varepsilon > 0$ and define

$$m_\varepsilon := \inf_{\|x\| = \varepsilon} V(x). \tag{57}$$

Because the $\varepsilon$-sphere $\mathcal{E} = \{x \in \mathbb{R}^n : \|x\| = \varepsilon\}$ is compact and $V$ is continuous, the infimum is achieved on $\mathcal{E}$. Positive-definiteness of $V$ further implies $m_\varepsilon > 0$.

Define

$$c := \frac{2(1 - \bar{\alpha})}{\underline{\sigma}} > 0.$$

Because $V$ is continuous at the origin and $V(0) = 0$, for every threshold $\eta > 0$ there exists a radius $\delta(\eta) > 0$ such that

$$\|x\| < \delta(\eta) \implies V(x) < \eta.$$

Choosing $\eta = m_\varepsilon/c$ and setting $\delta = \delta(m_\varepsilon/c)$, gives

$$\|x\| < \delta \implies V(x) < \frac{m_\varepsilon}{c}. \tag{58}$$

Now, pick any index $k$ with $x_k \in \mathcal{S}$ and $\|x_k\| < \delta$. From (58), we have the bound

$$V(x_k) < \frac{m_\varepsilon}{c}. \tag{59}$$

Using the two–step Lyapunov condition (56) and the fact that $\sigma_1, \sigma_2 \geq \underline{\sigma}$, we obtain

$$\underline{\sigma}(V(x_{k+1}) + V(x_{k+2})) \leq 2(1 - \bar{\alpha})V(x_k) \implies V(x_{k+i}) \leq c\, V(x_k), \quad i = 1, 2. \tag{60}$$

Combining (60) with (59) gives

$$V(x_{k+1}) < m_\varepsilon, \qquad V(x_{k+2}) < m_\varepsilon.$$

By definition of $m_\varepsilon$, $V(x) < m_\varepsilon$ implies $\|x\| < \varepsilon$; hence

$$\|x_{k+1}\| < \varepsilon, \qquad \|x_{k+2}\| < \varepsilon. \tag{61}$$

Equation (61) shows that, starting from any state inside the $\delta$-ball, the trajectory remains inside the prescribed $\varepsilon$-ball.

Moreover, due to the strict decrease enforced by (56), it is not possible for both $V(\mathbf{x}_{k+1}) \geq V(\mathbf{x}_k)$ and $V(\mathbf{x}_{k+2}) \geq V(\mathbf{x}_k)$ to hold simultaneously. Therefore, at least one of the two future states must

satisfy $V(\mathbf{x}_{k+i}) < V(\mathbf{x}_k)$ for $i \in \{1, 2\}$, hence the trajectory enters a strictly smaller sublevel set of $V$ within two steps, allowing the generalized Lyapunov condition to be re-applied.

We now distinguish two cases to formalize the re-application logic:

**Case 1:** $\mathbf{x}_{k+1} \in \mathcal{S}$. Then the generalized condition can be re-applied at $\mathbf{x}_{k+1}$, and since $V(\mathbf{x}_{k+1}) < \epsilon$, the next steps $\mathbf{x}_{k+2}, \mathbf{x}_{k+3}$ will also lie within the $\epsilon$-ball by repeating the same argument.

**Case 2:** $\mathbf{x}_{k+1} \notin \mathcal{S}$. Then $\mathbf{x}_{k+2} \in \mathcal{S}$, and again $V(\mathbf{x}_{k+2}) < \epsilon \Rightarrow \|\mathbf{x}_{k+2}\| < \epsilon$, allowing the condition to be re-applied at step $k + 2$.

**Conclusion.** In both cases, the trajectory remains within the $\epsilon$-ball for all time, and the generalized decrease condition continues to hold along the sequence. Therefore, for all $\epsilon > 0$, there exists $\delta > 0$ such that:

$$\|\mathbf{x}_0\| < \delta \quad \Rightarrow \quad \|\mathbf{x}_k\| < \epsilon \quad \forall k \geq 0, \tag{62}$$

proving Lyapunov stability of the origin relative to $\mathcal{S}$.

Therefore, we conclude that $\mathcal{S}$ is a valid inner approximation of the ROA. □

# F System Dynamics for Formal Verification

We describe below the continuous-time dynamics, control constraints, and verification domain $\mathcal{X}$ for each benchmark system used in our evaluations. All systems are discretized using explicit Euler integration with a time step of $\Delta t = 0.05$ s.

## F.1 Inverted Pendulum

The inverted pendulum is modeled as a single-link rigid body rotating about its base, subject to bounded torque control. The continuous-time dynamics are:

$$\dot{\theta} = \omega, \tag{63}$$

$$\dot{\omega} = \frac{g}{\ell} \sin(\theta) - \frac{\beta}{m\ell^2}\omega + \frac{u}{m\ell^2}, \tag{64}$$

where $\theta$ is the pendulum angle (upright at $\theta = 0$), $\omega$ is the angular velocity, and $u$ is the control input. We use the following parameters: gravity $g = 9.81$ m/s$^2$, mass $m = 0.15$ kg, length $\ell = 0.5$ m, and damping coefficient $\beta = 0.1$.

The control input is saturated to $u \in [-6, 6]$ Nm. The state vector is $\mathbf{x} = [\theta, \omega]^\top$, and the verification region is defined as:

$$\mathcal{X}_{\text{pendulum}} := \left\{ \mathbf{x} \in \mathbb{R}^2 \mid \theta \in [-12, 12], \ \omega \in [-12, 12] \right\}.$$

## F.2 Path Tracking

We consider a kinematic model for a ground vehicle tracking a circular path at constant forward speed $v$. The system state is $\mathbf{x} = [d_e, \theta_e]^\top$, where $d_e$ is the lateral distance to the reference path and $\theta_e$ is the heading error. The control input $u$ is the steering angle. The continuous-time dynamics are:

$$\dot{d}_e = v \sin(\theta_e), \tag{65}$$

$$\dot{\theta}_e = \frac{v}{L}u - \frac{\cos(\theta_e)}{R - d_e}, \tag{66}$$

where $L = 1$ m is the wheelbase length, $R = 10$ m is the radius of the reference path, and $v = 2$ m/s is the constant forward speed. The control input is bounded by $|u| \leq 0.84$.

The verification domain is defined as:

$$\mathcal{X}_{\text{path}} := \left\{ \mathbf{x} \in \mathbb{R}^2 \mid d_e \in [-3, 3], \ \theta_e \in [-3, 3] \right\}.$$

## F.3 2D Quadrotor

The 2D quadrotor models a rigid body with six states: horizontal and vertical positions $x, z$, pitch angle $\theta$, and their corresponding velocities $\dot{x}, \dot{z}, \dot{\theta}$. The system is controlled by two rotor thrusts

$u_1, u_2$, and the continuous-time dynamics are:

$$\ddot{x} = -\frac{1}{m}\sin(\theta)(u_1 + u_2), \tag{67}$$

$$\ddot{z} = \frac{1}{m}\cos(\theta)(u_1 + u_2) - g, \tag{68}$$

$$\ddot{\theta} = \frac{\ell}{I}(u_1 - u_2), \tag{69}$$

where $m = 0.486\,\text{kg}$, $\ell = 0.25\,\text{m}$, $I = 0.00383\,\text{kg}\cdot\text{m}^2$, and $g = 9.81\,\text{m/s}^2$. The state vector is $\mathbf{x} = [x, z, \theta, \dot{x}, \dot{z}, \dot{\theta}]^\top$. The control inputs are constrained to be within $[0, 2.5 \cdot u_{\text{eq}}]$, where $u_{\text{eq}} = \frac{mg}{2}$.

The verification domain is defined as:

$$\mathcal{X}_{\text{quadrotor}} := \left\{ \mathbf{x} \in \mathbb{R}^6 : \mathbf{x} \in [-0.75, 0.75]^2 \times \left[-\tfrac{\pi}{2}, \tfrac{\pi}{2}\right] \times [-4, 4]^2 \times [-3, 3] \right\}.$$

### F.4 Neural Network Architectures and Training Parameters

We adopt the same network structures used in Yang et al. [2024]. The configurations are summarized in Table 5. All networks use leaky ReLU activations.

Table 5: Neural network architecture for each system.

| System | Controller Network | Lyapunov Network |
|---|---|---|
| Inverted Pendulum | 4 layers, width 8 | 3 layers, widths [16, 16, 8] |
| Path Tracking | 4 layers, width 8 | Quadratic form |
| 2D Quadrotor | 2 layers, width 8 | Quadratic form |

