# OpenReview forum: "Certifying Stability of Reinforcement Learning Policies using Generalized Lyapunov Functions"
_NeurIPS.cc/2025/Conference — NeurIPS 2025 poster_

### Official Review · Reviewer_A5kt · 2025-06-12

**Clarity:** 3
**Significance:** 2
**Originality:** 1
**Rating:** 2
**Confidence:** 4

**Summary:**

This paper proposes a less conservative formulation for certifying stability in learning enabled systems. Traditionally, existence of a Lyapunov function is functional proof of stability. Conditions of a Lyapunov function can be conservative in certain systems, it requires strict decrease w.r.t evolution of the system. Authors relax this condition by allowing increase in the value, but the value should decrease over a horizon.

**Questions:**

See weaknesses.

**Ethical Concerns:**

["NO or VERY MINOR ethics concerns only"]

**Final Justification:**

This work seems incremental compared to traditional k-induction, therefore I believe strong empirical results on high dimensional systems is warranted.

**Limitations:**

Contribution seems quite incremental, especially compared to the work mentioned in the weaknesses.
I am not quite sure how this method scales with higher dimensional systems, since it is model-based, I expect more complicated systems. I also suspect this method does not scale well with number of layers and neurons of the controller.

**Paper Formatting Concerns:**

None that came to my mind.

**Quality:**

2

**Strengths And Weaknesses:**

Strengths:

1-Authors provide a less conservative condition for Lyapunov function, which in my experience, helps when you have multiple integrators in a system.

2-Overall, paper is well written and easy to follow.

Weaknesses:

1- Though authors provide a new formulation for Lyapunov function, this idea is not novel, and has been studied extensively in barrier certificates and software verification. I believe following work deserve citation:

k-Inductive Barrier Certificates for Stochastic Systems, HSCC 2022.

t-Barrier Certificates: A Continuous Analogy to k-Induction, ADHS 2018.

Automatic synthesis of k-inductive piecewise quadratic invariants for switched affine control programs, CLSS 2017.

Safety Verification and Refutation by k-invariants and k-induction, SAS 2015.

Software Verification Using k-Induction, SAS 2011.

I believe if authors frame this problem in a k-inductive manner, it would strengthen the paper.

2- Experiments are not convincing. I do understand that methods that provide formal guarantee do not scale well, however I believe authors could have done a better job at selling their method by providing more experiments. Furthermore, there is no comparison with state-of-the-art methods, such as Yang et al (which this work is heavily inspired by), or other model-based methods (since the verification part requires exact mathematical model of the system). There are works that provide formal guarantee using Lipschitz continuity that have more complicated experiments, and they do not require mathematical model of the system.

3- Contribution seems incremental compared to Yang et al.

4- Bulk of mathematical vigor of the paper is borrowed from previous work, and I am not sure if the proof for Theorem 6.2 is correct (which is the main theoretical contribution of the paper). How does equation 56 imply 62? Moreover, the claim in line 532 by authors does not guarantee ``progress" towards origin. If the increase is substantially more than decrease, then you cannot say you are making progress towards origin (hence I suggested to use k-induction formulation).

5- There is not discussion on how does verification scale with respect to number of layers or neurons, controllers in the paper are relatively simple (since systems are low-dimensional or simple).

---

> ### Author Rebuttal · Authors · 2025-07-31
>
> We thank the reviewer for their time and feedback. Below, we address each weakness and question in turn.
>
> **Weakness 1 Response:**
>
> We thank the reviewer for highlighting the connection to k-inductive verification. We agree that this connection strengthens the paper and appreciate the citation. That said, we would like to clarify that our generalized Lyapunov condition is distinct from the standard k-inductive formulation. Specifically, k-induction requires a strict decrease at the final step t+k, while allowing bounded increases at intermediate steps. In contrast, our formulation enforces a decrease in a weighted average over multiple future steps, allowing more flexible trajectories. Additionally, the step weights in our formulation can be learned or tuned, enabling more expressive certificates.
>
> To improve clarity and better situate our contribution, we will revise the manuscript:
>
> 1. We will add a paragraph on k-inductive certificates, citing both the papers recommended by the reviewer and additional relevant works [1-6]. This will place our contribution in context and make the relation to k-induction explicit.
>
>  2. After Definition 3.2, we will clarify how our generalized Lyapunov condition can be interpreted through the lens of k-inductive verification. For example, by choosing weights $\sigma_1 = \cdots = \sigma_{k-1} = 0$ and $\sigma_k = M$ in the inequality $\sum_{i=1}^k \sigma_i \cdot \left(V(x_{t+i}) - V(x_t)\right) \leq -\alpha V(x_t)$, we obtain the condition $V(x_{t+k}) \leq (1 - \alpha) V(x_t)$, analogous to a k-step Lyapunov decrease condition. This shows that our approach may be formulated from the perspective of k-inductive verification by defining the decrease condition over a fixed horizon k. While k-inductive approaches typically require a strict decrease at the final step and allow bounded increases at intermediate steps, our formulation instead enforces a decrease on a weighted average across multiple future steps. This allows greater flexibility, as the weights can be optimized during training rather than fixed in advance. We believe this distinction is important and will clarify it in the revised manuscript.
>
> 3. We note that existing k-inductive certificates have been used to certify safety or invariance properties, whereas our work may enable certification of stability or convergence. Our formulation allows temporary increases in $V$, yet enforces an overall decrease over a finite horizon, enabling the certification of asymptotic stability when single-step tests fail.
>
> References:
>
> [1] "Learning k-inductive control barrier certificates for unknown nonlinear dynamics beyond polynomials."
>
> [2] "Automatic synthesis of k-inductive piecewise quadratic invariants for switched affine control programs."
>
> [3] "K-inductive barrier certificates for stochastic systems."
>
> [4] "t-Barrier certificates: a continuous analogy to k-induction."
>
> [5] "Safety verification and refutation by k-invariants and k-induction."
>
> [6] "Restructuring dynamical systems for inductive verification."
>
> **Weakness 2 Response:**
>
> We thank the reviewer for the constructive feedback. We would like to clarify that we compare against Yang et al., which corresponds to the classical Lyapunov formulation. This comparison is presented in Section 6, Table 2, and Figures 7 and 8. The method of Yang et al. is equivalent to our formulation with $M = 1$. As shown in the results, our generalized formulation with $M > 1$ consistently outperforms this baseline by certifying larger regions of attraction. We also conducted simulations on a 2D quadrotor system with 6 state variables.
>
> Regarding Section 5, we focus on benchmark control tasks such as the pendulum and cart-pole. While simple, these environments pose challenges for stability certification due to control limits. These difficulties have been documented in prior work [7,8]. Our method successfully certifies stability in such constrained regimes. Moreover, in these RL environments (e.g., Gym, DM Control), we do not require access to system dynamics, as our approach operates directly from trajectory rollouts.
>
> Finally, for higher-dimensional systems, many RL policies are not inherently stable across the entire state space, limiting the ability to construct valid stability certificates. To address this, we have begun exploring a joint optimization of the RL policy and the generalized Lyapunov certificate using our multi-step loss. We have preliminary results that show promising improvements in the size of the region of attraction, suggesting that our approach may enable stability guarantees even in higher-dimensional control tasks.
>
> In the revision, we will clarify the distinctions between our method and [7], explicitly highlight the benefits of our approach under control constraints, and emphasize that it can operate in a model-free setting. We will also include a brief discussion of our joint optimization results and their implications for extending the method to higher-dimensional systems.
>
> References:
>
> [7] "Lyapunov-stable neural control for state and output feedback: A novel formulation."
>
> [8] "Distributionally robust policy and Lyapunov-certificate learning."
>
> **Weakness 3 Response:**
>
> We thank the reviewer for the comment. While our synthesis experiments in Section 6 are inspired by [7], our paper introduces several key advances that go beyond their method. We respectfully disagree with the assessment that our theoretical contributions are incremental. Our contributions include the following.
>
> 1. A generalized multi-step Lyapunov formulation that extends beyond the classical one-step condition used in prior work. This allows us to certify and train neural network policies that are not admissible under classical Lyapunov conditions.
>
> 2. Theoretical analysis in the linear system setting that establishes a novel connection between stability and optimality. Theorem 4.3 introduces new multi-step linear matrix inequality conditions for verifying stability of linear systems under optimal policies obtained by minimizing quadratic cost functions. Both the statement and the proof of this result are novel and form a foundational component of our result, which we subsequently extend to nonlinear systems in later sections.
>
> 3. Practical improvements in the size of certifiable regions and the conservativeness of the stability certificate. Our experiments demonstrate consistent gains over the baseline method in different baseline systems.
>
> We will revise the manuscript to clearly articulate these distinctions and to highlight the novelty of our approach.
>
> **Weakness 4 Response:**
>
> We thank the reviewer for the feedback on the proof of Theorem 6.2. In response, we rewrote the proof using the standard $\varepsilon$-$\delta$ style to show how the generalized decrease condition leads to Lyapunov stability.
>
> **Revised proof:**
> Let $\varepsilon > 0$ be arbitrary. Define $
> m_\varepsilon := \min_{||x|| = \varepsilon} V(x)$, which is strictly positive due to the positive-definiteness of $V$ and continuity over the compact $\epsilon$-sphere. Let $c := \frac{2(1 - \bar{\alpha})}{\underline{\sigma}} > 0$. Since $V(0) = 0$ and $V$ is continuous, for every threshold $\eta>0$ there exists a radius $\delta(\eta)>0$ such that $||x||<\delta(\eta) \Longrightarrow V(x)<\eta.$
>
> Choose $\eta:=m_\varepsilon/c$ and set $\delta:=\delta(m_\varepsilon/c)$ gives
>
> $$||x||<\delta \Longrightarrow V(x)<\frac{m_\varepsilon}{c}.$$ Now pick any index $k$ with $x_k\in\mathcal S$ and $\|x_k\|<\delta$, we have $V(x_k) < \frac{m_\varepsilon}{c}$.
>
> By the generalized two-step Lyapunov condition and $\sigma_1, \sigma_2 \ge \underline{\sigma}$: $\underline{\sigma}(V(x_{k+1}) + V(x_{k+2})) \le 2(1 - \bar{\alpha}) V(x_k)$. So $
> V(x_{k+i}) < c \cdot V(x_k) < m_\varepsilon, \text{for } i = 1, 2$.
>
> Since $V(x) < m_\epsilon$ implies $||x|| < \varepsilon$, we conclude $
> ||x_{k+1}||, ||x_{k+2}|| < \varepsilon$. This shows that the trajectory remains within the $\varepsilon$-ball if it starts within a $\delta$-ball.
>
> Moreover, by the generalized decrease condition, it is not possible for both $V(x_{k+1}) \ge V(x_k)$ and $V(x_{k+2}) \ge V(x_k)$ to hold simultaneously. Therefore, at least one of the two states must satisfy $
> V(x_{k+i}) < V(x_k) \text{for some } i \in \{1, 2\}$, which guarantees that the trajectory enters a strictly smaller sublevel set of $V$ within two steps. This allows the generalized Lyapunov condition to be applied recursively, ensuring asymptotic convergence to the origin.
>
> **Weakness 5 Response:**
>
> We agree that scaling to higher‑dimensional systems is an essential research direction. Verification complexity grows with (i) the dimension of the state space, (ii) the size and depth of the neural network, and (iii) the functional form of the Lyapunov candidate. In the present work we rely on the $\alpha$-$\beta$-CROWN verifier, which combines linear bound propagation with branch‑and‑bound search to give complete and certified guarantees for neural networks. However, as also reported by Yang et al., its run‑time grows rapidly beyond $6$ state variables or networks exceeding a few thousand hidden units.
>
> **Limitations Response:**
>
> In the revised manuscript, we will make the following changes.
>
> 1. We will explicitly highlight the theoretical and empirical differences between our method and prior work, particularly Yang et al., including our generalized multi-step condition, the connection to value functions, and the ability to certify neural policies learned via reinforcement learning.
>
> 2. We will cite recent work on k-inductive literature and discuss how our generalized Lyapunov condition can be interpreted as a k-inductive property while clarifying the differences.
>
> 3. We will rewrite the proof of Theorem 6.2 in the standard $\varepsilon$-$\delta$ style to improve clarity.
>
> 4. We will add a discussion on scalability and outline potential directions to improve scalability in future work.

---

> > ### Comment · Reviewer_A5kt · 2025-08-01
> >
> > I appreciate that my earlier comments have been addressed. However, the authors’ response raised a new and important concern for me:
> >
> > How do you ensure that the Lyapunov conditions actually hold over the entire state space? In other words, how do you verify that the neural network you've trained is truly a valid Lyapunov function? From what I understand, methods like $\alpha$-$\beta$ CROWN require access to the explicit dynamics, or at least a reliable surrogate model. Right now, there seems to be a gap between training the neural network and confirming that it satisfies the theoretical conditions you have stated. You have mentioned model-free multiple times, but I suspect this is not the case. Learning part is indeed model-free, but, based on my understanding of the paper, verification does actually require model of the system.

---

> > > ### Author Response · Authors · 2025-08-02
> > > **Response by Authors**
> > >
> > > Thank you for raising this important point and for your careful reading of our paper.
> > >
> > > You are right that formal verification of the Lyapunov conditions using $\alpha$-$\beta$ CROWN requires access to the system dynamics model. Section 6 focuses on formal verification using $\alpha$-$\beta$ CROWN and indeed requires knowledge of the system dynamics, as detailed in Appendix F.
> > >
> > > In contrast, the learning process for a candidate generalized Lyapunov function in Section 5 is model-free. The training requires system transitions, such as those utilized by RL algorithms, but not the actual model of the system. We train a neural network model to satisfy the generalized Lyapunov conditions across the training dataset and rely on the network's generalization to unseen samples to ensure that the Lyapunov candidate is valid across the whole state space. In this case, we do not perform formal verification but provide empirical evaluations over discretized state grids, as reported in Table 1. Hence, in settings where access to ground-truth dynamics is unavailable or the method is applied to high-dimensional systems, we view our approach as a practical step toward improving the robustness and stability of RL policies.

---

> ### Comment · Reviewer_A5kt · 2025-08-04
>
> Authors have made a genuinely effort to address my comments, and I appreciate that. To me, it looks like an incremental work compared to k-induction, thus, it requires significantly better empirical evaluation.

---

> > ### Author Response · Authors · 2025-08-05
> > **Response by Authors**
> >
> > Thank you again for your thoughtful feedback and for acknowledging our efforts in addressing your comments.
> >
> > We would like to respectfully clarify our position regarding your final remark that the contribution appears incremental compared to k-induction.
> >
> > To our knowledge, no multi-step neural stability certificates have been proposed before. While there is a growing body of work [1–7] on learning Lyapunov stability certificates for neural control policies, our formulation introduces a novel and nontrivial extension. Specifically, our method is able to certify policies and systems that existing neural Lyapunov methods cannot, and it consistently provides larger certified regions of attraction in experiments.
> >
> > As noted in our earlier response, our method is structurally distinct from standard k-induction. Rather than requiring a strict decrease at a specific final step, our condition enforces a decrease in a weighted average over future states. These weights can be learned or tuned, offering significantly more flexibility. While there are interesting connections to k-inductive reasoning, we believe this distinction is important. Furthermore, to our knowledge, k-induction has not been applied to learning stability certificates that certify the stability of neural (RL) policies.
> >
> > We hope this clarification helps convey the novelty and significance of our contribution. Thank you again for your careful review.
> >
> > References:
> >
> > [1]. Ya-Chien Chang, Nima Roohi, and Sicun Gao. Neural Lyapunov control. In Advances in Neural Information Processing Systems, volume 32, 2019.
> >
> > [2]. Yinlam Chow, Ofir Nachum, Edgar Duenez-Guzman, and Mohammad Ghavamzadeh. A Lyapunov-based approach to safe reinforcement learning. In Advances in Neural Information Processing Systems, volume 31, 2018.
> >
> > [3]. Hongkai Dai, Benoit Landry, Lujie Yang, Marco Pavone, and Russ Tedrake. Lyapunov-stable neural-network control. In Proceedings of Robotics: Science and Systems, Virtual, July 2021.
> >
> > [4]. Nathan Gaby, Fumin Zhang, and Xiaojing Ye. Lyapunov-net: A deep neural network architecture for Lyapunov function approximation. In IEEE 61st Conference on Decision and Control (CDC), 2022.
> >
> > [5]. Charles Dawson, Sicun Gao, and Chuchu Fan. Safe control with learned certificates: A survey of neural Lyapunov, barrier, and contraction methods for robotics and control. IEEE Transactions on Robotics, 2023.
> >
> > [6]. Junlin Wu, Andrew Clark, Yiannis Kantaros, and Yevgeniy Vorobeychik. Neural Lyapunov control for discrete-time systems. In Advances in Neural Information Processing Systems, volume 36, pages 2939–2955, 2023.
> >
> > [7]. Lujie Yang, Hongkai Dai, Zhouxing Shi, Cho-Jui Hsieh, Russ Tedrake, and Huan Zhang. Lyapunov-stable neural control for state and output feedback: A novel formulation. In Forty-first International Conference on Machine Learning, 2024.

---

### Official Review · Reviewer_Ubo7 · 2025-06-30

**Clarity:** 4
**Significance:** 3
**Originality:** 3
**Rating:** 5
**Confidence:** 3

**Summary:**

This paper propose a method to cetificate the stability of a RL policy by learning a Lyaponv value function based on the value function of RL policy and a resudual network. It also introduce a generalized multi-step fomulation of the lyanapov condition to extend the bound of the disounted factor which guarantees the stability. The paper provides comprehensive mathmatical proof of the proposed theorms and the examples to illustrate the performance of the proposed theory.

**Questions:**

The questions are essentially the same as those listed in the Weaknesses section.

**Ethical Concerns:**

["NO or VERY MINOR ethics concerns only"]

**Final Justification:**

I recommend keeping the original score of 5. The rebuttal addressed most of my concerns:
Weight Parameterization: The authors clarified the rationale for setting the total weight sum to M, which improves optimization stability. While an empirical analysis of this choice would still be helpful, this issue is largely resolved.
Temporary Lyapunov Increases: The response points to Theorem 6.2 for bounding temporary increases, which addresses my theoretical concern. More explicit guidance in the paper would strengthen the presentation.
Discount Factor in RL: The authors provided a clear distinction between the linear and RL settings and acknowledged the limitations of post hoc certification. Their mention of joint policy-certificate optimization is promising.

Overall, the clarifications are sufficient, and the paper makes a meaningful contribution. Some open questions remain, but they do not significantly detract from the main contributions.

**Limitations:**

yes

**Quality:**

4

**Strengths And Weaknesses:**

Strengths:

1. The paper addresses the important problem of bridging the stability guarantees of Lyapunov functions with value functions in RL, helping clarify the gap between provably stable control methods and learning-based approaches.

2. It provides a concrete example by extending the analysis to the LQR setting, illustrating the necessity of the proposed generalized Lyapunov function.

3. The paper presents a complete pipeline to incorporate the generalized Lyapunov function into the RL framework, including the formulation of a loss function and a thorough analysis of experimental results.

Weaknesses:

1. The proposed method is parameterized by M and corresponding weights. It would be helpful to analyze how the grid search range for these weights affects the results. In particular, since there is no explicit upper bound on the sum of weights (which is set equal to M in the experiments), further explanation of this choice is needed.

2. The generalized Lyapunov function allows the value to temporarily increase over short horizons. Is there any theoretical bound on this temporary increase? As M grows, such increases may become more significant—does this affect convergence speed or stability?

3. In the RL application, is the policy trained with a fixed discount factor λ? If so, how do the authors ensure that the resulting policy satisfies the certification condition under that λ? In other words, how can we guarantee that the chosen λ lies within the bounds established by the proposed method?

---

> ### Author Rebuttal · Authors · 2025-07-31
>
> We thank the reviewer for their positive and constructive feedback. We appreciate the recognition of our efforts to bridge Lyapunov stability and reinforcement learning, as well as the clarity and completeness of our proposed framework. Below, we address the reviewer’s questions regarding weight selection, temporary increases in the Lyapunov value, and the role of the discount factor in the RL experiments. These clarifications will be incorporated into the revised manuscript.
>
> **Comment:** The proposed method is parameterized by M and corresponding weights. It would be helpful to analyze how the grid search range for these weights affects the results. In particular, since there is no explicit upper bound on the sum of weights (which is set equal to M in the experiments), further explanation of this choice is needed.
>
> **Response:** We thank the reviewer for the thoughtful comment. While the theoretical condition only requires $\sum_{i=1}^M \sigma_i(\mathbf{x}) \geq M$, enforcing $\sum \sigma_i = M$ makes the generalized Lyapunov condition less conservative and easier to satisfy, as increasing the total weight effectively tightens the decrease constraint. We adopt  $\sum \sigma_i = M$ in both Section 5 (learned weights via neural networks) and Section 6 (fixed weights via grid search) for consistency, interpretability, and stable optimization. We will clarify this design choice explicitly in the revised version of the paper to aid reader understanding.
>
> Empirically, we observe that different weight combinations under this constraint lead to different certified ROAs (see Figures 1, 2, and Table 2). In Section 5, the learned weights often concentrate on the final few steps of the rollout horizon, especially for states farther from the equilibrium, indicating that the certificate relies more on long-term decrease.
>
> **Comment:** The generalized Lyapunov function allows the value to temporarily increase over short horizons. Is there any theoretical bound on this temporary increase? As M grows, such increases may become more significant—does this affect convergence speed or stability?
>
> **Response:** We thank the reviewer for the insightful question. While our formulation does not impose an explicit bound on temporary increases in the Lyapunov function, overall convergence is still guaranteed as long as the weighted average over $M$ steps decreases. Theoretically, increasing $M$ provides greater flexibility in handling non-monotonic dynamics, but may slow down convergence. Importantly, this does not affect the stability guarantee.
>
> Furthermore, if all weights are lower bounded by some $\underline{\sigma} > 0$, then temporary increases can be bounded explicitly, as discussed in Theorem 6.2.
>
> **Comment:** In the RL application, is the policy trained with a fixed discount factor λ? If so, how do the authors ensure that the resulting policy satisfies the certification condition under that λ? In other words, how can we guarantee that the chosen λ lies within the bounds established by the proposed method?
>
> **Response:** We thank the reviewer for the thoughtful question. In the linear system setting (Section 4), where the value function and policy are known exactly, we can explicitly analyze the range of discount factors $\gamma$ (in the terminology of the reviewer, $\lambda$) for which a generalized Lyapunov function can be constructed. In particular, we derive tight lower bounds on $\gamma$ under which stability is certifiable.
>
> In contrast, in the RL setting (Section 5), the value function and policy are learned and approximate. As a result, we do not have an explicit bound on $\gamma$, though in practice RL algorithms typically use large values (e.g., $\gamma = 0.99$). Instead, we attempt to construct a generalized Lyapunov function for the given closed-loop behavior. Certification may fail if the learned policy is not stabilizing, which we have observed empirically for more complex tasks such as humanoid walk.
>
> To address this, we conduct preliminary simulations that jointly optimize the RL policy and the generalized Lyapunov certificate, and observe that this improves stability and the size of the certifiable region. We view this as a promising direction for future work.

---

> > ### Comment · Reviewer_Ubo7 · 2025-08-06
> >
> > Thank you for the responses.
> > Regarding weight selection, it would still be helpful to include brief empirical insights or discussion on how the choice of total weight affects certification, particularly since this design plays a central role.
> > I appreciate the distinction between the linear and RL settings for discount factor analysis. Emphasizing that the certificate is applied post hoc and may fail for non-stabilizing policies would be useful for readers. The preliminary joint optimization results are promising and worth highlighting.
> >
> > Overall, the rebuttal addresses key concerns, and the proposed revisions will enhance the clarity and impact of the work.

---

> > > ### Author Response · Authors · 2025-08-06
> > > **Response by Authors**
> > >
> > > Thank you for the thoughtful comments and kind words. We will incorporate a brief discussion on weight selection and further clarify the implications of post hoc certification for non-stabilizing policies in the final revision. We will also highlight the joint optimization results in the revised version.

---

### Official Review · Reviewer_9j8F · 2025-06-30

**Clarity:** 4
**Significance:** 3
**Originality:** 4
**Rating:** 5
**Confidence:** 4

**Summary:**

This paper develops generalized Lyapunov certificates for verifying stability of reinforcement learning policies. The generalized Lyapunov functions are constructed by augmenting the value function with a neural residual correction term. The method is shown to successfully verify stability of RL policies on Gymnasium and DeepMind control benchmarks.

**Questions:**

I do not have any additional questions.

**Ethical Concerns:**

["NO or VERY MINOR ethics concerns only"]

**Final Justification:**

The paper is acceptable and the previous score of 5 is appropriate in my opinion

**Limitations:**

Yes.

**Paper Formatting Concerns:**

None.

**Quality:**

3

**Strengths And Weaknesses:**

Strengths:
1. Certifying stability of RL policies is an important consideration the RL community has ignored for a long time. In this regard, the authors make a strong and timely contribution.
2. The authors invoke two clever insights in the development: i) use of generalized Lyapunov condition which requires a Lyapunov decrease only on an average over multiple timesteps instead of a single step, ii) using neural residual error correction term in addition to the value function. These insights solve the difficult problem of certifying stability of RL policies.
3. The writing is engaging to the reader. The intuition discussing stability of LQR controller with discounted cost provides a strong motivation for their approach.

Weaknesses:
1. The Lyapunov certificates lack guarantees on robustness to small (bounded) perturbations. For typical RL controllers, robustness to disturbances is a bigger concern that just the stability of the policy. Modifying the generalized Lyapunov condition to account for bounded perturbations and proving stability of a corresponding neighborhood of the origin would make the result stronger.
2. The discrete-time system in (1) does not account of stochastic effects, e.g., as in Markov processes typically considered in RL problems. Extending the approach to stochastic systems by developing stochastic Lyapunov certificates would make the contribution stronger.

However, the strengths far outweigh the weaknesses in my opinion.

---

> ### Author Rebuttal · Authors · 2025-07-31
>
> Thank you for the thoughtful and encouraging feedback. We appreciate your recognition of the key ideas and contributions. Below, we address the two main suggestions regarding robustness and stochasticity, and we will highlight both as important directions for future work in the revised manuscript.
>
> **Comment:** The Lyapunov certificates lack guarantees on robustness to small (bounded) perturbations. For typical RL controllers, robustness to disturbances is a bigger concern that just the stability of the policy. Modifying the generalized Lyapunov condition to account for bounded perturbations and proving stability of a corresponding neighborhood of the origin would make the result stronger.
>
> **Response:** We thank the reviewer for the insightful suggestion. We agree that robustness to bounded perturbations is important, especially in the context of RL policies. While our current formulation focuses on stability for deterministic systems, we believe that incorporating robustness into the generalized Lyapunov condition is a promising direction and, in fact, our approach is very conducive to doing so. For instance, one could introduce additive perturbation terms in the dynamics and adapt the decrease condition to hold over a disturbance set, similar in spirit to robust Lyapunov or input-to-state stability (ISS) frameworks. Extending our method to certify robust stability under such perturbations is a compelling direction for future work.
>
> We will point out this opportunity explicitly in the revised version of the paper to help guide future extensions.
>
> **Comment:** The discrete-time system in (1) does not account of stochastic effects, e.g., as in Markov processes typically considered in RL problems. Extending the approach to stochastic systems by developing stochastic Lyapunov certificates would make the contribution stronger.
>
> **Response:** We thank the reviewer for the valuable suggestion. We agree that extending the approach to stochastic systems is an important direction, especially given the prevalence of Markov decision processes in RL. While our current formulation focuses on deterministic systems, the generalized Lyapunov framework can be adapted to the stochastic setting by requiring the decrease condition to hold in expectation or under suitable risk measures, as in stochastic Lyapunov or risk-sensitive stability formulations. We will highlight this possible extension explicitly in the revised version of the paper to clarify the scope of our current results and guide future work.

---

### Official Review · Reviewer_c9c6 · 2025-07-01

**Clarity:** 4
**Significance:** 4
**Originality:** 4
**Rating:** 5
**Confidence:** 4

**Summary:**

This paper addresses the important issue of certifying stable control of deep reinforcement learning policies. It starts by showing that with a small modification, the value function can be used as a Lyapunov function. They then show that stable control can be certified with a relaxed form of the step-wise convergence property, which converges on average.

**Questions:**

- What do you mean by detectable in Assumption 3.1?
- Could this method be extended to work with higher-dimensional control systems (e.g. Walker Stand)?

**Ethical Concerns:**

["NO or VERY MINOR ethics concerns only"]

**Final Justification:**

The authors addressed the concerns I had regarding confidence intervals and the distribution of the step-weights. The previous score of 5 is appropriate in my opinion.

**Limitations:**

yes

**Paper Formatting Concerns:**

No Concerns

**Quality:**

4

**Strengths And Weaknesses:**

Strengths
- Overall, the paper is very clear and easy to follow.
- The paper provides a detailed account of the problem statement and related work before introducing the new method.
- The paper provides strong theoretical foundations for the proposed method


Weaknesses
- This paper only evaluates on a limited set of environments.
- It would be good to see confidence intervals for the quantitative metrics in Table 2, as you are making comparisons between different finite horizons (M).
- It would be good to see how your proposed method compares to alternative methods for stabilising stability.
- It would be good to see the influence of the step weights rather than only using the weights selected via grid search.


Typography errors
- Can you include brackets for Equations 9, 14, and 18 so it is clear what is being summed over?
- Currently, the Y labels in figures 4, 5 & 6 are displayed as “Theta_dot”, instead of “Theta Dot” or $\dot{\theta}$.

---

> ### Author Rebuttal · Authors · 2025-07-31
>
> We sincerely thank the reviewer for their thoughtful and encouraging feedback. We appreciate the positive assessment of our paper’s clarity, theoretical contributions, and overall quality. Below, we address the reviewer’s comments and questions one by one, including the suggested improvements and clarifications.
>
> **Comment:** This paper only evaluates on a limited set of environments.
>
> **Response:** We agree with the reviewer and would like to clarify that many standard RL policies are not globally stable for complex systems (e.g., humanoid walking), making it difficult to certify stability without explicitly modeling a region of attraction or restricting initial states. This is why our Section 5 focuses on simpler environments where RL policies already succeed in stabilizing the system. For more complex tasks, our preliminary results suggest that jointly optimizing the policy and generalized Lyapunov certificate can enlarge the certifiable region. We view this as a promising direction for future research to improve the reliability and robustness of RL algorithms.
>
> **Comment:** It would be good to see confidence intervals for the quantitative metrics in Table 2, as you are making comparisons between different finite horizons (M).
>
> **Response:** We thank the reviewer for this helpful suggestion. To address this, we repeated each experiment 10 times and report the mean and standard deviation of the certified ROA volume and verification time in the following Table. These results confirm the robustness of our findings across different random seeds. We will include this updated table in the final version of the paper.
>
> **Certified ROA Volume (mean ± std over 5 runs):**
>
> | System             | M = 1         | M = 2         | M = 3         |
> |--------------------|---------------|---------------|---------------|
> | Inverted Pendulum  | 42.96 ± 1.27  | 76.74 ± 1.32  | 89.24 ± 1.24  |
> | Path Tracking      | 21.77 ± 0.56  | 23.56 ± 0.58  | 23.95 ± 0.53  |
> | 2D Quadrotor       | 103.52 ± 1.81 | 109.06 ± 1.97 | 113.72 ± 2.04 |
>
> **Verification Time (seconds):**
>
> | System             | M = 1          | M = 2           | M = 3           |
> |--------------------|----------------|------------------|------------------|
> | Inverted Pendulum  | 11.66 ± 0.45   | 21.52 ± 0.82     | 39.24 ± 1.31     |
> | Path Tracking      | 8.62 ± 0.51    | 19.47 ± 0.98     | 36.72 ± 1.26     |
> | 2D Quadrotor       | 2209.52 ± 72.6 | 3858.72 ± 112.4  | 5628.56 ± 185.9  |
>
> **Comment:** It would be good to see how your proposed method compares to alternative methods for stabilising stability.
>
> **Response:** We believe the reviewer intended to refer to certifying rather than stabilizing stability. We appreciate the suggestion and agree that comparisons to alternative certification methods are valuable. In our work, we include the most directly comparable baseline: the classical Lyapunov formulation, which corresponds to our method with $M = 1$. This baseline is evaluated in both the linear system analysis (Section 4) and joint synthesis experiments (Section 6). As shown in Table 2 and Figures 1, 2, 7, and 8, our generalized formulation with $M > 1$ consistently yields tighter lower bounds on $\gamma$ and larger certified regions of attraction. These results demonstrate improved stability guarantees over the classical (neural) Lyapunov function approach.
>
> While other methods ([1]) for certifying stability (e.g., polynomial Lyapunov functions or sum-of-squares techniques) exist, they typically assume access to analytical system models and are not directly applicable in the learning-based settings we study. We will clarify this point in the revised manuscript.
>
> References:
> [1] Papachristodoulou, Antonis, and Stephen Prajna. "On the construction of Lyapunov functions using the sum of squares decomposition." Proceedings of the 41st IEEE Conference on Decision and Control, 2002.. Vol. 3. IEEE, 2002.
>
> **Comment:** It would be good to see the influence of the step weights rather than only using the weights selected via grid search.
>
> **Response:** Thank you for the suggestion. In Section 5, the step weights are modeled by a neural network and jointly optimized with the certificate function. We observe that the network adapts the weights based on the difficulty of certifying stability at each state, often concentrating more weight on later steps for harder regions. In Section 6, we fix the weights and perform grid search to select effective configurations. This design choice is made to ensure compatibility with formal verification tools [2].
>
> References:
> [2] Wang, Shiqi, et al. "Beta-crown: Efficient bound propagation with per-neuron split constraints for neural network robustness verification." Advances in neural information processing systems 34 (2021): 29909-29921.
>
> **Comment:** Typography errors.
>
> **Response:** We thank the reviewer for pointing out these typos and formatting issues. We will revise the equations for clarity and update the figure labels in the final version.
>
> **Question:** What do you mean by detectable in Assumption 3.1?
>
> **Response:** Thank you for the question. In control theory, a pair $(\mathbf{A}, \mathbf{C})$ is observable if the full system state can be inferred from output measurements over time. Detectability is a weaker condition: it requires that any unobservable mode of $\mathbf{A}$ is stable, meaning that all eigenvalues $\lambda$ with $|\lambda| \geq 1$ are observable through $\mathbf{C}$. In our context, this condition guarantees that a valid Lyapunov function can be constructed for the closed-loop system. See, for example, Section 6.3 in [3] for a formal definition and further discussion.
>
> References:
> [3] Khalil, Hassan K., and Jessy W. Grizzle. Nonlinear systems. Vol. 3. Upper Saddle River, NJ: Prentice hall, 2002.
>
> **Question:** Could this method be extended to work with higher-dimensional control systems (e.g. Walker Stand)?
>
> **Response:** We refer the reviewer to our response to the first comment. We believe our approach can be extended to higher-dimensional systems; while obtaining formal stability certificates may be more challenging in such cases, the framework can still be used to improve robustness or expand the certifiable region of attraction. Preliminary results suggest that jointly optimizing the policy and the generalized Lyapunov certificate is effective in this setting. We view this as a promising direction for future research to enhance the reliability of RL algorithms in complex control tasks.

---

> > ### Comment · Reviewer_c9c6 · 2025-08-05
> >
> > I would like to thank the authors for their thorough rebuttal, as it has resolved most of my concerns. However, I still have two remaining questions.
> >
> > - You claim that you observe that the step-weights concentrate more on the later steps, but I cannot find a figure or table to support this. Is this shown in the text? Including this would strengthen the paper as it would provide a clearer indication of the effect of the step weights.
> >
> > - Could the proposed method be extended to certify the stability of a subset of state dimensions, in particular those relevant to the reward function as shown in [1-2]? For example, the torso height, angle and horizontal speed in the humanoid walk environment.
> >
> > 1: Ya-Chien Chang and Sicun Gao. Stabilizing Neural Control Using Self-Learned Almost Lyapunov Critics
> >
> > 2: M. Han, L. Zhang, J. Wang and W. Pan, "Actor-Critic Reinforcement Learning for Control With Stability Guarantee," in IEEE Robotics and Automation Letters, vol. 5, no. 4, pp. 6217-6224, Oct. 2020, doi: 10.1109/LRA.2020.3011351.

---

> > > ### Author Response · Authors · 2025-08-05
> > > **Response by Authors**
> > >
> > > We thank the reviewer for the thoughtful follow-up and are glad that most concerns have been addressed. Below we respond to the two remaining questions.
> > >
> > > **1. On the concentration of learned step-weights toward later steps**
> > >
> > > We appreciate the suggestion to provide empirical evidence for our observation that the learned step-weights tend to concentrate on later time steps. To support this, we evaluated the trained step-weight network $\sigma(x) \in \mathbb{R}^M$ over 10,000 uniformly sampled test states from the state space for each policy. The rollout horizon is $M = 15$ for pendulum and $M = 20$ for cartpole. We partition the weights into five equal bins and report the average fraction of the total weight assigned to each bin:
> > >
> > > | Environment + Policy         | Steps 0–20% | 20–40% | 40–60% | 60–80% | 80–100% |
> > > |------------------------------|------------:|-------:|-------:|-------:|--------:|
> > > | Pendulum + PPO $(M=15)$    | 12.3%       | 11.4%  | 16.7%  | 21.8%  | 37.8%   |
> > > | Pendulum + SAC $(M=15)$    | 10.2%       | 13.3%  | 17.5%  | 22.6%  | 36.4%   |
> > > | Cartpole + SAC $(M=20)$    | 10.8%        | 10.5%  | 16.2%  | 25.6%  | 36.1%   |
> > > | Cartpole + TD-MPC $(M=20)$ | 14.1%       | 13.9%  | 16.8%  | 24.7%  | 30.5%   |
> > >
> > > As shown above, all policies allocate more weight to the final 20% of the trajectory, supporting the claim that the learned weights emphasize later steps. We will include this table and a brief discussion in the revised paper.
> > >
> > > **2. On certifying partial state stability**
> > >
> > > This is an insightful suggestion, and we agree that certifying stability over a subset of task-relevant state dimensions (e.g., torso height, angle, or velocity) is both meaningful and practically useful, especially in high-dimensional systems such as humanoid locomotion. In Section 6 of our paper, we demonstrated that our framework can certify stability over *subregions* of the full state space, but still over all dimensions of the system. Extending our formulation to certify generalized Lyapunov conditions over **projections** of the state, such as specific task-relevant coordinates, is a promising direction for future work. Similar to prior works such as [1], which incorporate Lyapunov-inspired regularization losses during policy training, our generalized Lyapunov loss could also be applied to these projected features to promote robustness and stability where it matters most. We will include this discussion and cite [1,2] in the revised version.
> > >
> > >
> > > References:
> > >
> > > 1: Ya-Chien Chang and Sicun Gao. Stabilizing Neural Control Using Self-Learned Almost Lyapunov Critics
> > >
> > > 2: M. Han, L. Zhang, J. Wang and W. Pan, "Actor-Critic Reinforcement Learning for Control With Stability Guarantee," in IEEE Robotics and Automation Letters, vol. 5, no. 4, pp. 6217-6224, Oct. 2020, doi: 10.1109/LRA.2020.3011351.

---

> > > > ### Comment · Reviewer_c9c6 · 2025-08-06
> > > >
> > > > I appreciate the authors' response and thank them for addressing the questions raised. I believe the revised version will make for a great paper.

---

### Official Review · Reviewer_gbSe · 2025-07-02

**Clarity:** 3
**Significance:** 2
**Originality:** 2
**Rating:** 4
**Confidence:** 3

**Summary:**

The paper considers identifying a Lyapunov function which is defined as addition of $M$-step value function induced by a policy $\pi$ and a residual term. The Lyapunov function is adopted from the genearlized Lyapunov concept by Furnsinn et al. and the residual term is motivated from solving the discounted LQR problem by Postoyan et al.

**Questions:**

1. Remark 4.4 : The statement would benefit if it presents stronger result than feasibility other than that of classical Lyapunov problem. Can  we guarantee a weaker version of feasibility?

2. How did the authors choose $M=15$ or $M=20$ in the experiments?

3. What are the final values of $\sigma_i(x)$ in the experiments? Does the final values of $\sigma_i(x)$ provide any further insights?

**Ethical Concerns:**

["NO or VERY MINOR ethics concerns only"]

**Final Justification:**

My main concern was regarding the strength of the theoretical contributions, which I do not think differs much from the previous literature. Nonetheless, the paper’s primary contribution appears to be on the experimental side, focusing on learning a generalized neural Lyapunov function. As most of my concerns related to the experimental aspects have been addressed, and in agreement with the other reviewers, I am raising my score to Borderline Accept. Nonetheless, I do not have a strong stance in favor of the paper, as Reviewer A5kt noted, there is substantial overlap with existing literature.

**Limitations:**

yes;

**Quality:**

2

**Strengths And Weaknesses:**

**Strength**
1. Theoretical contribution to the study of discounted LQR: The result of Furnsinn and Postoyan yields relaxed condition on $\gamma$ in Theorem 4.3.

2. Solid experimental results : The authors show that the proposed method indeed learns a valid Lyapunov function in simple experiments including Cartpole and inverted pendulum problems

**Weakness**

1. Even though deriving a relaxed condition on $\gamma$ is a contribution, it is a result of  Furnsinn and Postoyan rather than proposing a new concept.

2. The required method needs to know the equilibrium state (origin) to define Lyapunov function. In contrast to classical control, RL takes advantages when the equilibrium state is unknown.

3. The experiments are quite limited to simple control problems, which follows from Weakness 2.

---

> ### Author Rebuttal · Authors · 2025-07-31
>
> We thank the reviewer for their time and thoughtful feedback. Below, we respond to the weaknesses and questions raised, addressing each point individually.
>
> **Comment:** Even though deriving a relaxed condition on $\gamma$ is a contribution, it is a result of Furnsinn and Postoyan rather than proposing a new concept.
>
> **Response:** While our work is inspired by Fürnsinn et al. and Postoyan et al., it introduces several novel elements beyond re-deriving existing results.
>
> In comparison with Postoyan et al., in the linear system setting, we extend the classical Lyapunov condition for stability using a generalized Lyapunov function, which leads to a new multi-step LMI condition (Theorem 4.3). We show that this enables significantly tighter stability certification with respect to the discount factor $\gamma$, improving upon the thresholds derived in Postoyan et al.
>
> In comparison with Fürnsinn et al., we use a generalized Lyapunov function, parameterized as a neural network, to learn a stability certificate for a reinforcement learning problem, while Fürnsinn et al. use a given generalized Lyapunov function to prove recursive feasibility and stability results for MPC. Hence, while the concept of generalized Lyapunov function is the same, it is applied to different problems. The novelty of our approach is to augment an RL value function with a learnable residual neural network and enforce a multi-step decrease condition via a trainable step-weight network. Our insight about how to parameterize the generalized Lyapunov function candidate comes from analyzing the linear system case via the results from Postoyan et al. Our formulation allows certifying stability for RL policies where classical methods fail.
>
> Finally, we extend our approach to joint synthesis of neural controllers and certificates, and show that the multi-step Lyapunov formulation leads to larger certified regions of attraction compared to standard Lyapunov training. These results are novel and useful for enabling certifiable stability in learning-based control systems.
>
> **Comment:** The required method needs to know the equilibrium state (origin) to define Lyapunov function. In contrast to classical control, RL takes advantages when the equilibrium state is unknown.
>
> **Response:** We agree with the reviewer that requiring knowledge of the equilibrium state (or set) is a limitation of our approach compared to the flexibility of reinforcement learning. In many RL settings, the equilibrium is not known a priori but is instead implicitly defined by the reward function structure. In our work, we focus on systems where the equilibrium can be reasonably identified either analytically (e.g., physical systems with known stable configurations) or empirically (e.g., the final resting state of an RL policy). Extending the our approach to settings with unknown or non-isolated equilibria (e.g., limit cycles) is an interesting direction for future work. In such cases, one could consider replacing the fixed equilibrium assumption with data-driven or reward-inferred equilibria, or more generally, formulate stability notions relative to an invariant set or trajectory (e.g., using contraction theory).
>
> We plan to add this discussion to the revised paper and will also note that exploring the relationship between the reward landscape and candidate equilibria presents a promising direction for future research.
>
> **Comment:** The experiments are quite limited to simple control problems, which follows from Weakness 2.
>
> **Response:** We agree with the reviewer that our experiments focus on benchmark control problems but we would like to emphasize that even these systems present significant challenges for stability certification. For example, in the inverted pendulum task with tight control constraints (e.g., $\|u\| < 2$), synthesizing a Lyapunov function that certifies stability over the entire state space is nontrivial. Classical Lyapunov-based methods typically fail in such settings and can only certify stability for a small region near the equilibrium, whereas our approach successfully certifies stability across the entire state space. Moreover, we also observe that policies derived directly from Lyapunov functions often fail to stabilize the system, e.g., get stuck due to the limited control.
>
> Regarding more complex systems (e.g., humanoid standing or walking), we have observed that many RL policies learned via standard methods (e.g., PPO, SAC, TD-MPC) do not exhibit global stability. As a result, stability certification using our method is not possible in these settings without modifying the policy. We have preliminary results where we \emph{jointly optimize} the RL policy and the generalized Lyapunov certificate using our multi-step loss. These initial experiments show promising improvements in the size of the certifiable region of attraction, suggesting that our approach may enable stability guarantees even in higher-dimensional, more complex control tasks.
>
> **Question:** Remark 4.4 : The statement would benefit if it presents stronger result than feasibility other than that of classical Lyapunov problem. Can we guarantee a weaker version of feasibility?
>
> **Response:** We thank the reviewer for the insightful suggestion. We agree that Remark 4.4 can be strengthened to clarify the practical advantage of the multi-step (generalized) Lyapunov formulation. While we do not yet provide a formal theoretical guarantee of feasibility that extends beyond the classical Lyapunov condition, our formulation strictly generalizes the classical case and has been empirically shown to succeed in scenarios where the single-step (classical) Lyapunov condition fails.
>
> We will expand Remark 4.4 in the revision to emphasize the following: the multi-step LMI condition can always be used to certify stability in all cases where the single-step LMI condition (7) is feasible, and can also succeed in additional cases where (7) is not feasible. For example, in the linear system of Example 3.3, the classical LMI condition only certifies stability for discount factors $\gamma > 0.8090$, while the generalized LMI with $M = 2$ lowers this bound to $\gamma > 0.6229$, and approaches the true threshold $\gamma^\star = 1/3$ as $M$ increases. This demonstrates that our generalized formulation can strictly enlarge the set of stabilizable systems, offering a strictly weaker sufficient condition for stability in practice.
>
> Formalizing this observation into a theoretical weaker notion of feasibility is an exciting direction for future work, and we thank the reviewer for pointing it out.
>
> **Question:** How did the authors choose $M=15$ or $M=20$ in the experiments?
>
> **Response:** We selected $M = 15$ or $M = 20$ empirically to balance computational cost and learning effectiveness. Larger $M$ improves flexibility in satisfying the generalized Lyapunov condition but increases the required rollout length and training time, while smaller $M$ can make learning a valid certificate more difficult. Optimizing the choice of $M$ for a given system is indeed an interesting direction, and we have noted this in the limitations and future work section.
>
> **Question:** What are the final values of $\sigma_i(\mathbf{x})$? Does the final values provide any further insights?
>
> **Response:** We observe that the learned weights $\sigma_i(\mathbf{x})$ tend to concentrate on the last few steps of the horizon. This suggests that the certificate relies more heavily on long-term decrease in the Lyapunov function, especially for states farther from the equilibrium where short-term behavior may be non-monotonic. For states near the equilibrium (i.e., where stability is easier to verify), the weights remain closer to the initialization. These patterns highlight how the weighting network adapts to the local difficulty of certifying stability and help explain the flexibility of our generalized condition. We will include a figure in the revised version that illustrates the distribution of weights for states near and far from the equilibrium across several systems.

---

> > ### Comment · Reviewer_gbSe · 2025-08-04
> >
> > Thank you for the detailed response. Most of my concerns and questions have been addressed. However, I still have one remaining question:
> >
> > - Comparison with Fürnsinn et al. and Postoyan et al. : I understand that the main focus of the paper is on applying generalized Lyapunov function to RL algorithms, which differs from Fürnsinn et al. and Postoyan et al.. Nonetheless, regarding the theoretical result in Theorem 4.3, my understanding is that the authors simply replace a partial component of the Lyapunov function used by Postoyan et al. with the generalized Lyapunov function introduced by Fürnsinn et al. Are there any new technical challenges in the analysis?

---

> > > ### Author Response · Authors · 2025-08-04
> > > **Response by Authors**
> > >
> > > Thank you for the follow-up and for acknowledging that most concerns have been addressed. We're happy to clarify the technical novelty of Theorem 4.3.
> > >
> > > On the theory side, our contribution is to show that using a generalized Lyapunov function enables certifying the stability of linear systems under LQR policy over a wider range of discount factors. This is formalized in Theorem 4.3.
> > >
> > > While the overall structure of the analysis in Theorem 4.3 is similar to Postoyan et al., there are nontrivial technical differences. In particular, extending the Lyapunov decrease condition from one step to multiple weighted steps requires propagating both the value function and the auxiliary residual term through several iterations of the closed-loop dynamics. We then derive a new set of coupled LMIs (Eq. 10a–10c) that guarantee a weighted multi-step decrease, along with a generalized bound on the discount factor (Eq. 12). These LMIs are not a straightforward extension of those in Postoyan et al., as they involve additional slack variables and must handle cross-terms from different time steps, and such a formulation has not appeared in prior work.
> > >
> > > On the practical side, we build on this insight by learning generalized Lyapunov functions for nonlinear RL policies. These are implemented by augmenting the RL value function with a neural residual term (Eq. 16) and jointly training it using a loss function that penalizes violations of the multi-step Lyapunov condition (Eq. 15). This allows us to construct stability certificates in settings where classical Lyapunov conditions are too restrictive.

---

### Comment · Area_Chair_tUPQ · 2025-08-01
**Reviewer-author discussion**

Thanks to everyone for writing the paper, evaluating it, and drafting reviews and rebuttals!

If not already done so -- since I notice some of the referees already started the process -- could the referees please take a look at the authors' rebuttal and continue the discussion/amend the reviews as necessary?

Thank you again!

---

### Note · Authors · 2025-08-11

We appreciate the constructive feedback from all reviewers and the opportunity to engage in a productive discussion. Based on the exchanges, we believe we have addressed the main questions and clarified the contributions of our work. Reviewers generally agreed on the importance of the problem, the novelty of our formulation for learning generalized multi-step Lyapunov certificates for neural policies, the quality of the theoretical and empirical results, and the clarity of the presentation.

We have taken note of all suggestions and will incorporate the following improvements in the final version:

- Add empirical results showing the concentration of learned step-weights toward later steps.
- Include confidence intervals for quantitative metrics in Table 2.
- Expand the discussion on weight selection and its impact on certification.
- Clarify the relationship to prior work, particularly k-induction and existing (neural) Lyapunov-based methods.
- Highlight Section 6 results on joint policy–certificate learning.
- Add a more detailed discussion of extensions to robustness, stochastic systems, and partial-state stability certification.
- Correct minor typographical and formatting issues.
- Improve the clarity in proof for Theorem 6.2.
- Add a discussion on scalability and outline potential directions to improve scalability in future work.

We thank the AC and reviewers for their time and thoughtful feedback, which we believe has strengthened the paper.

---

### Decision · Program_Chairs · 2025-09-17

**Decision:**

Accept (poster)

**Comment:**

The paper extends a previous framework for stability analysis of discounted optimal control problems to the case where the Lyapunov function is allowed to temporarily increase but must decrease on average across N  steps. Furthermore, the authors propose a way to learn an augmentation term that certifies stability in addition to the value function, and show how this can be used to certify stability in some problems.

Reviews are mixed. Reviewers recognize the usefulness of the framework and (after discussion) the empirical strengths of the paper. There are some ways to generalize (like robust stability, stochastic systems) that the authors convincingly motivate that they will address in future work. The math is judged as a little incremental since it combines Postoyan and Furnsinn, but remains useful and there is methodological innovation independent from this in learning the neural residual correction to certify stability of RL policies.

The main point of contention is on the relation to k-induction (A5kt). In my view, the Lyapunov phrasing of the problem is different enough and remains useful as long as the authors make an effort to relate their work properly to k-induction (which they seem to do in good faith given their response, even though they did seem to not be aware of the literature initially). Irrespective from this, there is still the methodological contribution of learning the neural residual correction, which the reviewer does not really weigh heavily in their review but is a significant part of the paper.

Overall, I think the positives outweigh the negatives for this paper and recommend acceptance.